# spinDrop: a droplet microfluidic platform to maximise single-cell sequencing information content

Joachim De Jonghe [1,2,10], Tomasz S. Kaminski [1,3,10], David B. Morse[4], Marcin Tabaka[5,6], Anna L. Ellermann [1], Timo N. Kohler [1], Gianluca Amadei[7], Charlotte E. Handford[7], Gregory M. Findlay [2], Magdalena Zernicka-Goetz [7,8], Sarah A. Teichmann [9] & Florian Hollfelder [1] ✉

Droplet microfluidic methods have massively increased the throughput of single-cell sequencing campaigns. The benefit of scale-up is, however, accompanied by increased background noise when processing challenging samples and the overall RNA capture efficiency is lower. These drawbacks stem from the lack of strategies to enrich for high-quality material or specific cell types at the moment of cell encapsulation and the absence of implementable multi-step enzymatic processes that increase capture. Here we alleviate both bottlenecks using fluorescence-activated droplet sorting to enrich for droplets that contain single viable cells, intact nuclei, fixed cells or target cell types and use reagent addition to droplets by picoinjection to perform multi-step lysis and reverse transcription. Our methodology increases gene detection rates fivefold, while reducing background noise by up to half. We harness these properties to deliver a high-quality molecular atlas of mouse brain development, despite starting with highly damaged input material, and provide an atlas of nascent RNA transcription during mouse organogenesis. Our method is broadly applicable to other droplet-based workflows to deliver sensitive and accurate single-cell profiling at a reduced cost.

Droplet microfluidic methods have fundamentally transformed the field of single-cell RNA-sequencing (scRNA-seq) by increasing the number of cells that can be profiled in a single experiment by more than an order of magnitude compared to plate-based assays[1]. These technological advances have propelled the field into the age of molecular atlases that aim to resolve the full spectrum of cellular heterogeneity across entire organs[2–8] or organisms[9,10]. Although combinatorial indexing methods, such as sci-RNA-seq3[11] surpass droplet microfluidic methods in terms of throughput[1], most atlases to date have been generated using the commercial 10x Chromium platform[3], which can be explained by earlier adoption and extensive standardisation of commercial kits and analysis software[12]. Although the popularity of droplet-based approaches for single-cell profiling is evident, some fundamental challenges associated with the methodology remain to be resolved.

[1]Department of Biochemistry, University of Cambridge, Cambridge, United Kingdom. [2]Francis Crick Institute, London, United Kingdom. [3]Department of Molecular Biology, Institute of Biochemistry, Faculty of Biology, University of Warsaw, Warsaw, Poland. [4]Department of Chemistry, University of Cambridge, Cambridge, United Kingdom. [5]International Centre for Translational Eye Research, Warsaw, Poland. [6]Institute of Physical Chemistry, Polish Academy of Sciences, Warsaw, Poland. [7]Department of Physiology, Development and Neuroscience, University of Cambridge, Cambridge, United Kingdom. [8]California Institute of Technology, Division of Biology and Biological Engineering, Pasadena, USA. [9]Wellcome Sanger Institute, Wellcome Genome Campus, Hinxton, United Kingdom. [10]These authors contributed equally: Joachim De Jonghe, Tomasz S. Kaminski. ✉e-mail: fh111@cam.ac.uk

The vast majority of single-cell RNA sequencing campaigns will aim to maximise the number of cells profiled and the number of unique cDNA molecules that can be confidently detected per cell, sometimes referred to as sensitivity[13], to yield statistical power to downstream analyses such as differential gene expression analysis. However, there is currently a trade-off between the cost of library preparation and gene detection rates per cell. Although the commercial 10x Chromium[12] outclasses open-source protocols such as inDrop[14] and Drop-seq[15] in terms of sensitivity (2.5- and 1.2-fold higher gene detection rates, respectively)[16], the associated library preparation cost per cell remains prohibitive for large-scale profiling (twofold higher than respective open-source methods). Therefore, molecular atlasing experiments would hugely benefit from a method with reduced cost per cell while maintaining high sensitivity to derive meaningful biological conclusions.

Furthermore, the quality of the data derived from droplet-based methodologies can suffer from artefacts that may confound data analysis: RNA released from lysed cells indiscriminately enters into droplet compartments and generates a contaminating readout that is not cell-specific anymore, contributing both to cost and compromising data interpretation[17–19]. Cell debris or damaged cells can also be captured in those experiments, further complicating the identification of live cells in a dataset. Although experimental safeguards can be implemented to mitigate these effects, such as pre-sorting of live cells using fluorescence-activated (FACS) or magnetic-activated cell sorting (MACS), they do not guarantee viability at the moment of encapsulation. On the contrary, shear stress during flow cytometry or long processing times may lead to altered transcriptomes[20] or cell death[21]. Live cell enrichment procedures typically require large amounts of input material[22] and do not remove contaminating sequences, such as primer dimers and concatemers, generated from empty droplets. Similarly, bioinformatic tools (EmptyDrops[18], SoupX[17], emptyNN[23] and DropletQC[19]) have been employed to remove confounding effects from empty droplets and/or low-quality material. However, the performance of these approaches depends on sample quality and the cell types profiled. Finally, cell multiplets resulting from cell co-encapsulation or aggregation may further compromise data analysis by generating artificial cell populations. Although tools have been developed for their identification during data processing[24,25], they mostly resolve heterotypic multiplets (i.e. multiplets containing cells from different cell types), and do not remove the burden of sequencing costs associated with these artefacts. To this end, a generalisable methodology to extract droplets containing single viable cells, intact nuclei or specific cell types from the pool of empty droplets, droplets containing cell debris, multiplets or unwanted cell types would reduce cost and remove confounding artefacts found in droplet microfluidic datasets.

To alleviate the aforementioned bottlenecks, we have developed sorting picoinjection inDrop (spinDrop), a scalable droplet microfluidic method that delivers highly-sensitive 3′ mRNA sequencing of single viable cells, intact nuclei, paraformaldehyde-fixed samples or target cell types at a reduced cost. The protocol first employs fluorescence-activated droplet sorting (FADS)[26] to exclusively extract target material, followed by a picoinjection step[27] in which an improved reverse transcription formulation is added. We demonstrate five-fold higher gene detection rates compared to inDrop[14] to match the resolution obtained with the 10x Chromium platform[12], while significantly reducing the noise linked to empty droplets and poor quality cells. We demonstrate the utility of our workflow by profiling mouse brain development using a damaged sample as input, while maintaining high data output quality. The multi-step capabilities of our workflow to power high-throughput '-omics' applications were demonstrated by profiling nascent RNA transcription during mouse organogenesis at E8.5 using scEU-seq implemented in droplets.

## Results

### Overview of the spinDrop workflow

The spinDrop microfluidic workflow aims to generate highly-sensitive single-cell RNA-seq libraries from small quantities of input material with minimal contamination from damaged cells and empty droplets. To achieve this, a multi-step droplet microfluidic workflow was established (Fig. 1A). First, a strategy was devised to alleviate the current bottlenecks of pre-sorting for viability using FACS or MACS, which necessitate long processing times, large input materials and may damage the input material further due to mechanical shearing. To this end, we implemented in-line FADS[26] to enrich for droplets that match a sorting *criterion* (e.g. reporting on cell viability) after cell co-encapsulation with barcoded microgels in water-in-oil emulsions. Cells are first stained using a viability dye (Calcein-acetoxymethyl (AM)). For sorting of fixed cells or intact nuclei, a DNA stain can be employed (Vybrant Green) instead. For cell-type specific enrichment, fluorescently labelled antibodies targeting cell surface markers are used. The cells are then channelled in a flow-focusing device in conjunction with barcoded polyacrylamide microgels (with an inDrop v3 barcoding scheme[28,29]) and a lysis mixture (Fig. 1B). Single-cells are hereby co-encapsulated with the microgels in water-in-oil emulsions and can be sorted further down in the microfluidic device. Cells trapped in a droplet are excited by an adjacent blue-laser optical emission fibre. If cells are viable at the moment of encapsulation, the acetoxymethyl group linked to the Calcein fluorophore (in the case of live cell sorting) is released by intracellular esterases[30], enabling green fluorescence emission via blue-light excitation. Fluorescence is then collected in a second optical fibre, which relays the information to a field-programmable gate array which controls the generation of an alternative current (AC) square signal by a set of electronic instruments (pulse generator, function generator and amplifier - Supplementary Fig. 1). If the detected levels of fluorescence exceed the signal generated from empty droplets or droplets containing damaged cells, an electrical pulse is triggered and delivered to the microfluidic sorting junction via the activation of electrodes filled with a 5 M NaCl solution[31]. The activation of electrodes charges the droplets via dielectrophoresis and diverts them from the lower hydrodynamic resistance channel (negative channel) to pull them towards the positive channel (Fig. 1C, Supplementary Figs. 1A,D, 2A, Supplementary Movie 1) for downstream processing.

Second, the sensitivity bottleneck of open-source platforms, in particular inDrop[14] upon which this workflow is based on, was addressed using multi-step enzymatic and incubation processing. Currently, droplet microfluidic single-cell protocols rely on a single enzymatic treatment at fixed temperatures to perform reverse transcription[12,14,15,32]due to incompatibility of proteolytic lysis and enzymatic reverse transcription. This limitation prevents the enhanced RNA capture rates observed in plate-based assays, which, for example, use proteinase K and heat denaturation during the lysis step to increase denaturation of the nuclear envelope, release nucleic acids from protein complexes and unfold secondary structures to boost RNA capture[33–37]. We previously described a droplet microfluidic methodology that enables multi-step reactions for single-cell sequencing to be carried out in droplets[37], employing a previously described microfluidic design termed 'picoinjector'[27](Supplementary Fig. 1B, C). We, therefore, sought to use this method to decouple cell lysis from reverse transcription (RT) by injecting a RT reaction mixture optimised for 3′ mRNA capture consecutively to cell lysis, to match workflows from the more sensitive plate-based assays. To achieve this, water-in-oil emulsions containing the sorted single-cell lysate are loaded into the picoinjector and spaced using fluorosurfactant oil. When approaching a junction with an electrode, the droplet interface is electrically disrupted and the pressurised flow of mixture is injected into the incoming droplet (Fig. 1D, Supplementary Movie 2). The droplets can then be further collected and incubated for reverse

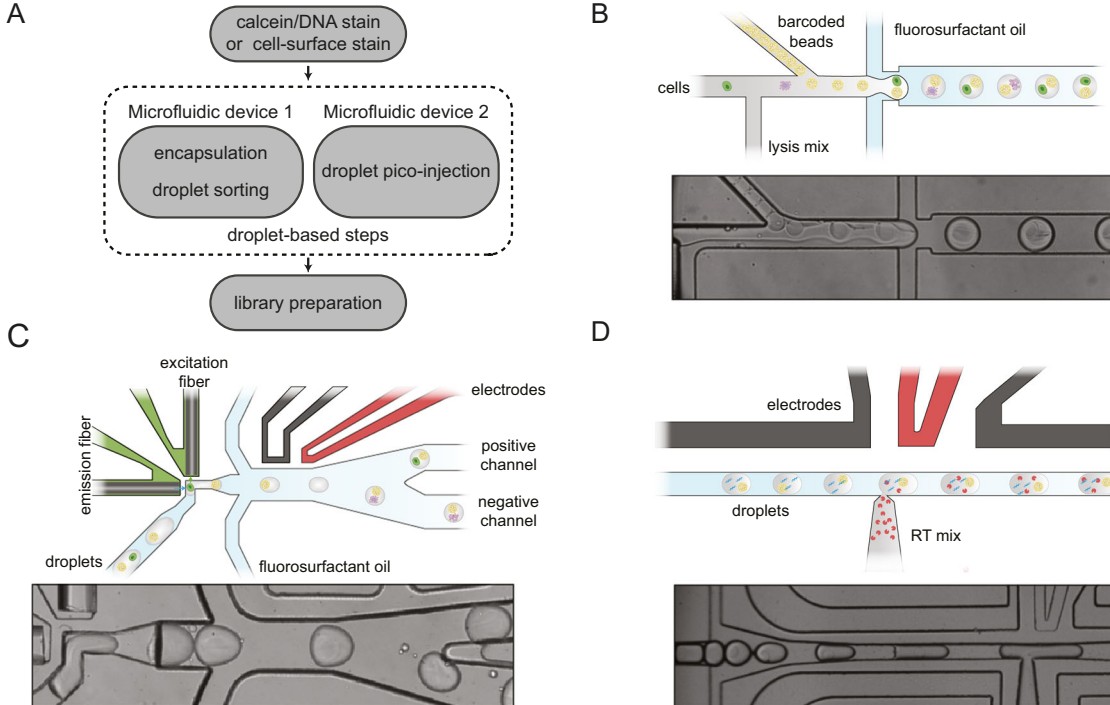

**Fig. 1 | Overview of the modular droplet microfluidic workflow for spinDrop ('sorting and picoinjection inDrop'). A** Schematic of the different steps to generate sequencing-ready libraries. First, intact whole cells are made detectable with a Calcein-AM viability stain, cell-surface marker stain, and intact nuclei using a DNA staining dye (such as the Vybrant DNA stain). Then the cells or nuclei are co-encapsulated with barcoded polyacrylamide beads, and droplets with viable cells, intact nuclei or specific cell types are enriched after encapsulation using in-line fluorescence-activated droplet sorting (FADS), hereby discarding empty droplets or droplets containing damaged material. The droplets containing the viable cells or intact nuclei are then lysed and heat-treated, and further re-injected in a second

microfluidic device, which will inject the reverse transcriptase mix using coalescence-activated picoinjection. **B** Schematic of the cell/nuclei and polyacrylamide bead droplet co-encapsulation process in the first microfluidic device (*top*). Brightfield image of the co-encapsulation process (*bottom*). **C** Schematic of fluorescence-activated sorting of droplets containing viable cells or intact nuclei from the pool of empty droplets or droplets containing damaged or unwanted material (*top*). Brightfield image of the in-line sorting process (*bottom*). **D** Schematic of the coalescence-activated droplet picoinjection of a reverse transcriptase mix (*top*). Brightfield image of the picoinjection process (*bottom*).

transcription before de-emulsification and downstream library preparation, following the inDrop protocol[14].

## FADS reduces background noise and increases cell loading in droplets beyond Poisson distribution

To characterise the capabilities of FADS to enrich for droplets containing single cells, HEK293T cells were stained with Calcein-AM, encapsulated without lysis reagents and sorted using a threshold that separated the background signal from the cellular fluorescence signal (Fig. 2A, Supplementary Fig. 2B) . The resulting sorted droplets were analysed on a fluorescence microscope, showing a stark enrichment (96.1%, n = 51) for droplets containing single viable cells (Fig. 2B). To further quantify the potential of our system to extract single viable cells from a challenging sample containing a large proportion of damaged cells, the input population was modified to incorporate a 1:1 ratio of dead and alive HEK293T cells treated with a dual green/red-live/dead stain (Calcein-AM and ethidium homodimer-1 respectively). To induce cell death, a concentration of 0.25% (w/v) IGEPAL CA-630 was added to half of the HEK293T cells which were incubated on ice for 15 minutes. Sorting of a 1:1 mixed population of dead and living cells showed a marked 19-fold enrichment for viable cells from the pool of droplets containing cells assessed for viability using a fluorescence microscope. 84.8% of the droplets contained a single viable cell, which surpasses the predicted value of 4.52% without sorting (Fig. 2C, Supplementary Data 1). On the other hand, the remainder of the droplets collected in the negative channel consisted almost exclusively of empty droplets (93%), with some dead cells (3.5%) and living cells (3.1%) (Fig. 2C). To assess whether discarding cellular multiplets as

a result of stochastic co-encapsulation is possible, an upper threshold on the voltage corresponding to the intensity of the fluorescence signal (3 V) was applied to a sample with a fivefold higher Poisson loading (defined by λ value which is the mean number of cells per droplet) of 1:1 mixed dead and living cells, to exclude brighter droplets that may contain multiple cells. Under these conditions, 30.3% of sampled droplets are predicted to contain a single cell (compared to 9.0% at λ = 0.1), which represents a threefold increase in processing throughput and may assist large-scale sampling endeavours, but comes at the cost of higher multiplet rates. At λ = 0.5, 77.2% of the sorted droplets contained a single viable cell after sorting, which largely outclasses the predicted values (Fig. 2C). While viable cells can be detected, some residual dead cells that do not emit fluorescence will statistically be co-encapsulated with viable cells and cannot be counterselected against using FADS. The proportion of droplets containing one dead and one living cell in the sorted population was 5.6%, slightly higher than the predicted value of 3.8%. This limit is due to stochastic co-encapsulation of low viability/low fluorescence cells, a process that is aggravated at high loading concentrations when isolating cells from a low viability pool .

To further determine whether discarding empty droplets from the analysis would lead to a lower fraction of background reads generated from empty droplets, a species-mixing experiment with human HEK293T and mouse embryonic stem cells (mESCs) was performed (under inDrop reaction conditions), with and without sorting. The proportion of reads matching to a cell barcode (through the computation of a coverage inflection point) was 58.0% for inDrop without sorting, similarly to previously reported values[14] and 88.4% for inDrop

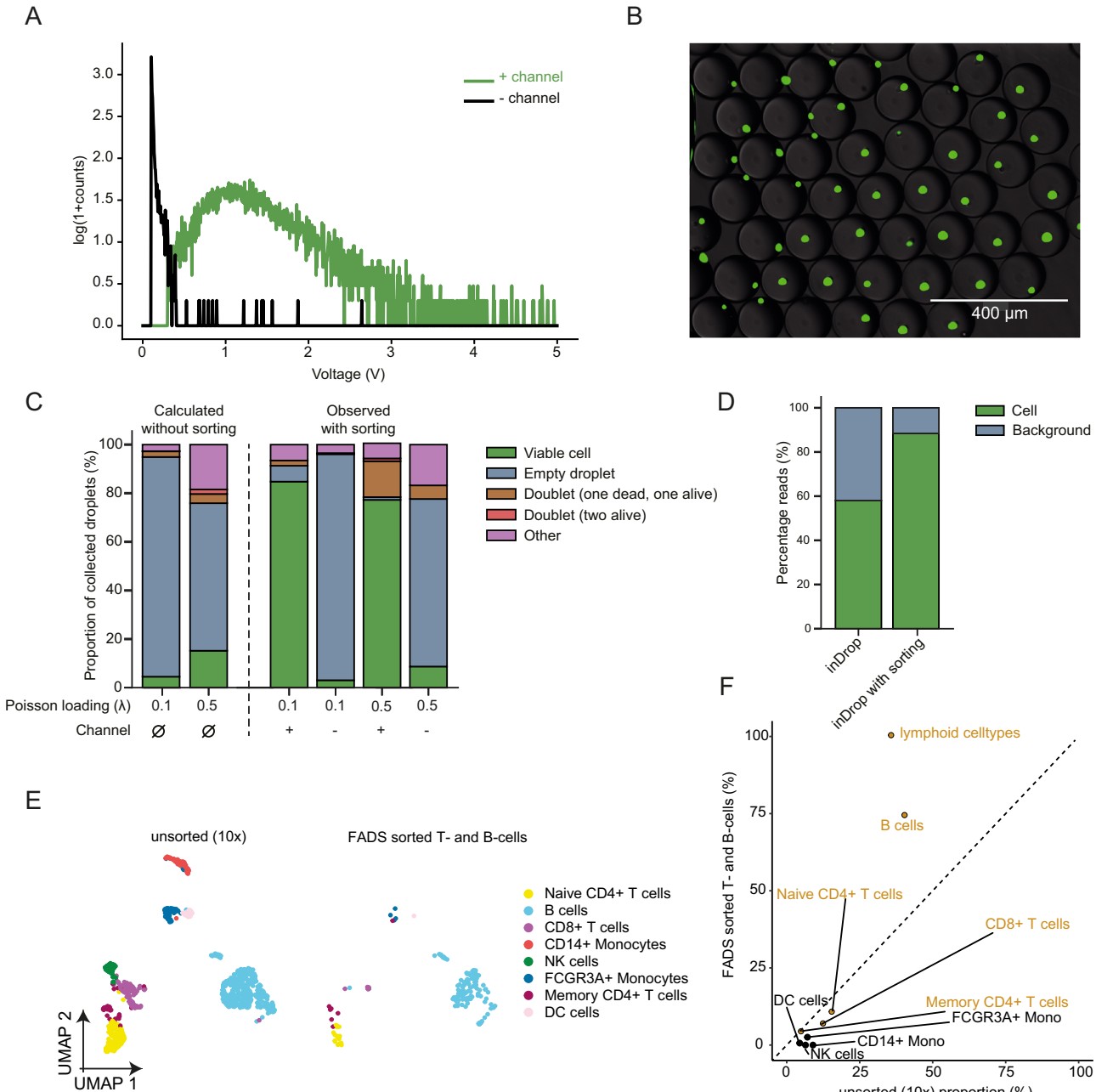

**Fig. 2 | Sorting of viable cells using FADS decreases background noise from cell-free RNA and dead cells. A** Measured signal of HEK293T cells stained with Calcein-AM using a detection fibre and photomultiplier tube, with thresholding on the background population (black) and the droplets with an encapsulated viable cell (green). Source data are provided as a Source Data file. **B** Overlay of fluorescence and brightfield images of the positive sorted population of droplets, showing the accurate single compartmentalisation of viable cells. Scale bar: 400 µm. **C** Sorting statistics using a dual live/dead stain on a 1:1 dead/alive population of HEK293T cells. Poissonian loading stands for a cell loading of λ = 0.1 and super-loading stands for a cell loading of λ = 0.5. (+) indicates counting on the sorted droplets, and (−) indicates counting on the unsorted droplets. The predicted values without sorting are on the left side of the dashed line, the observed values are on the right side of the dashed line. Source data are provided as a Source Data file. **D** Percentage of reads that are mapped to background, consisting mainly of primer concatemers, ambient RNA and degraded cells or cell debris; and cell barcodes for inDrop and sinDrop (inDrop with sorting), determined using the filtering statistics from the zUMIs pipeline[44]. **E** UMAP dimensional reduction plot showing the shared embedding for mouse PBMCs sequenced using the 10x Chromium v2 protocol and sequenced after on-chip FADS sorting using CD45R, CD19 and IgM as cell-surface specific fluorescent antibodies. **F** Proportional distribution of each cell type in the dataset in (**E**), illustrating an enrichment for lymphoid B- and T- cells in the FADS sorted dataset. The dotted line represents equal proportions across both datasets.

with sorting (Fig. 2D), documenting a clear gain in cellular read coverage for cultured cells that translates into lower sequencing costs and higher accuracy for data interpretation.

To further test the ability of our system to enrich for specific cell types using FADS, mouse peripheral blood mononuclear cells (PBMCs) were stained using phycoerythrin (PE) labelled antibodies specific to IGM, CD19 and CD45R and processed for sequencing (Supplementary

Fig. 2C). Projecting the dataset on a UMAP embedding containing an unsorted reference dataset generated using the 10x Chromium revealed a marked depletion of myeloid cell types (Fig. 2E, Supplementary Fig. 2D, E), and an overall 1.8-fold enrichment in B-cell content. Monocytes, such as CD14+ monocytes, were entirely depleted, whereas they accounted for 9% of the population without enrichment (Fig. 2F). To verify that our sorting parameters did not significantly

affect the population of B-cells (mainly due to their size), the cell-cycle phase was profiled for all datasets, revealing that the proportion of cells in each phase was broadly similar between all datasets. This observation refutes that selections were based on cell sizes as the latter varies significantly throughout cell-cycle stages (Supplementary Fig. 2F).

These findings illustrate the benefits associated with in-line droplet sorting using FADS: to reduce cost and decrease potential confounding effects from empty droplets and degraded input material. Additionally, cell-type specific enrichment may further decrease cost and enable higher coverage of rare cell populations, in applications where FACS is incompatible, because of low cell numbers or viability.

**Improved lysis and reverse transcription reaction conditions increase RNA capture**

We next sought to improve RNA capture efficiency in our assay, which is a current bottleneck in open-source workflows such as inDrop[14]. In existing droplet-based single-cell transcriptomics approaches, including the 10x Chromium system, the content of the droplets cannot be modulated after encapsulation. This precludes implementation of efficient cell lysis before reverse transcription, as with best-in-class plate-based assays that use heat denaturation and proteinase K treatments to increase RNA yields[33,34], which are fundamentally incompatible with reverse transcription. We therefore sought to decouple both steps by performing lysis at the encapsulation stage followed by reverse transcription after picoinjection of the droplets containing the cell lysate. We tested several reaction conditions using HEK293T cells: 1) the addition of 7.5% (w/v) PEG 8000 as a crowding agent to increase sensitivity (as observed in the plate-based mcSCRB-seq protocol[34]); 2), the use of proteinase K and heat denaturation (70 °C for 10 minutes) in addition to the non-ionic detergent IGEPAL CA-630 used in the inDrop protocol; 3) different reverse transcriptases: Maxima H- and Superscript III RTs used in mcSCRB-seq[34] and Smart-seq3[33], respectively. Downsampling the dataset to 20,000 reads per cell across conditions revealed a gene detection rate of 1016 genes for the original inDrop reaction conditions, in line with previous findings[14,16]. Adding 7.5% (w/v) PEG8000 and decoupling lysis from RT resulted in a median of 3384 (with PEG) and 3403 (without PEG) detected genes per cell, demonstrating that molecular crowding does not yield higher sensitivity in droplet formats. However, performing lysis with a proteinase K and heat denaturation treatment increased gene detection rates more than threefold compared to inDrop, leading to an overall detection of 4005 genes using the Maxima reverse transcriptase and 4926 using Superscript III (Fig. 3A). Because the latter conditions demonstrated superior performance, the remainder of the manuscript will utilise this set of reaction conditions (dubbed 'spinDrop'). The gain in the number of genes detected per cell is accompanied by higher detection-levels of intronic reads, which amount to 36.8% for spinDrop (Supplementary Fig. 3A) compared to 31.8% for inDrop, which may benefit RNA velocity analyses[38]. To determine spinDrop's ability to increase the RNA velocity resolution, scVelo[39] was run on the samples in dynamic mode. As expected, the core transcriptional dynamics captured by velocity in the homogeneous HEK293T cultured cells were associated with cell cycle genes. The top 5 most dynamical genes for spinDrop were G2M markers which showed clear induction, steady-state and repression phases, similar to the 10x Chromium data (Supplementary Fig. 4A, B). This was less apparent in the inDrop data, which displayed most of the counts along the "spliced" axis, underlining that increased intronic coverage and sensitivity yielded superior dynamical modelling using spinDrop.

These improvements were then measured against the most sensitive droplet microfluidic platform[12], the 10x Chromium, by downsampling read cellular coverage from 5000 to 25,000 reads per cell to evaluate gene detection saturation. The downsampling curves demonstrated slightly lower sensitivity for the 10x Chromium platform compared to spinDrop, with a median of 4665 genes detected using 10x Chromium at 20,000 reads per cell (Fig. 3B). Higher median percentage of UMIs mapping to mitochondrial genes was obtained using spinDrop (10.6% for 10x Chromium v2, 9.3% for inDrop and 14.0% for spinDrop), further underlining protocol-specific capture (Supplementary Fig. 3B). A Wilcoxon rank sum test further established the core differences in the genes detected between the 10x Chromium and the spinDrop methodology. The analysis revealed that a total of 690 genes were significantly and robustly differentially expressed throughout the dataset (absolute values for log2 fold change > 1 and Bonferroni adjusted p-values < $10^{-5}$). Further annotation of the genes by biotypes showed an enrichment for non-coding RNAs and pseudogenes for spinDrop and some protein-coding genes in the 10x Chromium dataset (Supplementary Fig. 3D, Supplementary Data 2). From the list of significantly differentially expressed genes, spinDrop showed an enrichment of 291 genes which, classified proportionally per biotype, were mainly: 1) 2.1% lncRNAs, 2) 26.4% processed pseudogenes, 3) 63.9% protein-coding, 4) 2.7% transcribed processed pseudogenes and 5) 2.7% unprocessed pseudogenes. The 10x dataset, on the other hand, had 399 genes that were upregulated which, classified proportionally per biotype, were: 1) 99% protein coding genes and 2) 1% of lncRNAs. This illustrates some of the core differences between both methods, likely owing to differences in library preparation methods, such as the cDNA amplification process and lysis method. For example, spinDrop uses in vitro transcription for cDNA amplification and 10x Chromium uses PCR for cDNA amplification, which has been reported to yield different gene capture[40]. Another fundamental difference relies on the release of nuclei-localised RNA molecules such as lncRNAs, which could be found at higher levels in the spinDrop methodology, likely owing to more efficient lysis of the nuclear envelope due to the proteinase K and heat denaturation steps. Technology-specific preferential enrichment of gene classes may further inform mechanisms. For example, some of the enriched protein coding genes for the spinDrop method relate to Gene Ontology terms GO:0048024 (regulation of mRNA splicing, via spliceosome, FDR 2.0 $e^{-8}$) and GO:0043484 (regulation of RNA splicing, FDR 7.6 $e^{-11}$), hinting towards better definition of the splicing machinery in this methodology (Supplementary Fig. 3E). In addition, the spinDrop method enables capture of molecular spike-ins, which is prohibitively expensive in traditional droplet set-ups due to the overwhelming majority of droplets being empty. For example RNA spike-in controls developed by the External RNA Controls Consortium (ERCC)[41], which can be utilised to assist data normalisation, were readily captured using spinDrop without the sequencing cost burden associated with empty droplet capture (Supplementary Fig. 3E).

To test the reproducibility of spinDrop, two independent libraries prepared using HEK293T cells as an input were sequenced and analysed, showing similar gene and UMI capture per cell (Supplementary Fig. 3F, G). The two replicates homogeneously spread across the two first principal components during dimensional reduction with no library-specific bias (Fig. 2C) and correlation analysis of the average expression per gene showed high inter-replicate homology ($R^2 = 0.98$, Fig. 2D).

To verify that the spinDrop method reliably compartmentalised single-cells and preserved the compartment throughout the picoinjection step, a species mixing assay comprising mESCs and human HEK293T cells was performed in both cellular and nuclei formats (Fig. 3E). The analysis revealed a doublet rate of 2.9% for the cellular sample (Supplementary Fig. 4C) and 6.1% for the nuclei sample (Supplementary Fig. 4D), illustrating a low proportion of doublets and droplet merging throughout the microfluidic process. Finally, we used spinDrop to process paraformaldehyde-fixed HEK293T cells. Cell fixation is the method of choice for archiving clinical samples[42] and may position our methodology as uniquely suited to interrogate previously inaccessible sample types from

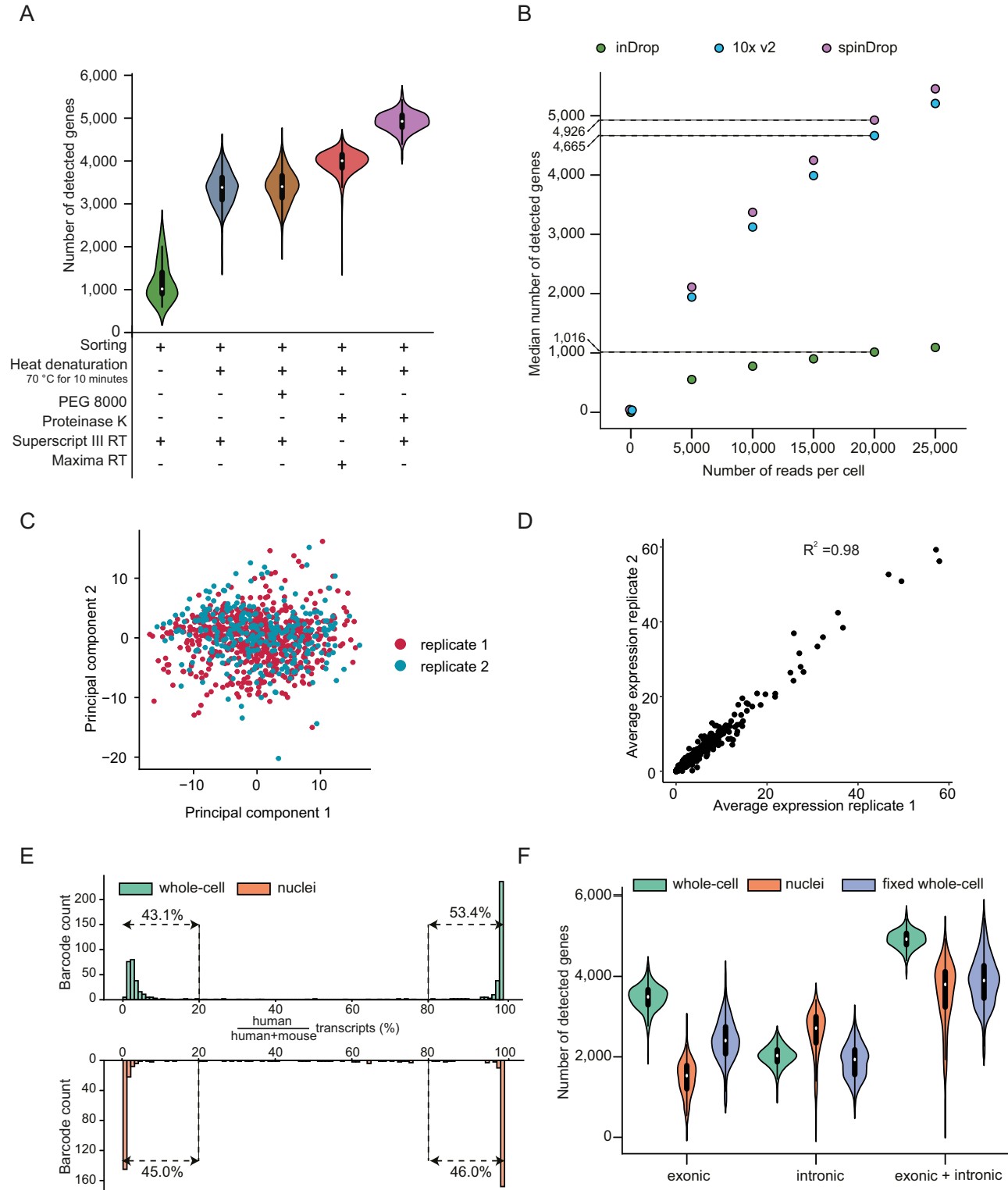

repositories. Contrasting all three formulations showed that nuclei had higher numbers of reads mapping to intronic regions (median of 2714 genes) compared to exonic regions (median of 1533 genes). This distribution was reversed for the whole cell samples, which detected a median of 2030 genes with reads mapping to introns and 3493 genes with reads mapping to exons. The sample with fixed cells displayed slightly lower gene detection rates, with 1934 genes with reads mapping to introns and 2404 genes with reads mapping to exons (Fig. 3F). High capture rates for both cytoplasmic and nuclear RNA molecules from fixed samples will broadly expand the number of single-cell methods directly applicable in a high-throughput format, and permit single-cell sequencing after storage, which will be beneficial to clinical samples. In contrast to the newly released fixed-RNA profiling 10x Chromium kits, spinDrop does not rely on probes for sequencing fixed cells, which should expand the number of species that can be investigated, and yield functional information (e.g. splicing or genotyping) to phenotyping experiments.

**Fig. 3 | Improved lysis procedures and picoinjection of reverse transcriptase mixture increase RNA capture efficiency. A** Gene detection rates for different reaction conditions using Calcein-AM stained HEK293T cells. Each dataset was downsampled to 20,000 reads per cell to allow for direct comparison between reaction conditions. The purple colour delineates the conditions for sorting picoinjection inDrop (spinDrop) used in the remainder of the manuscript. Heat denaturation was performed at 70 °C for 10 minutes. The white dot represents the median of the distribution, the thicker black bar represents the interquartiles range, the thin black bar represents data that extends to 1.5x the interquartiles range. **B** Gene detection downsampling curves comparing the median number of genes detected for HEK293T cells using the 10x Chromium v2, inDrop and spinDrop reaction conditions. The biological read length for each sample was cut to 61 bp to allow for a direct comparison between all datasets. The dashed line indicates median gene detection rates at 20,000 reads per cell. **C** PCA dimensionality reduction plot for both HEK293T replicates across principal components (PC) 1 and 2. **D** Correlation between the average expression for all genes in replicate 1 versus replicate 2. **E** Species mixing barcode proportional counts using mouse embryonic stem cells and HEK293T cells (green) or nuclei (orange) as an input for the spinDrop protocol. **F** Intronic and exonic UMI counts for HEK293T cells either prepared as whole cells (green), nuclei (orange) or fixed whole cells (purple), illustrating high capture across different input formats. The white dot represents the median of the distribution, the thicker black bar represents the interquartiles range, and the thin black bar represents data that extends to 1.5x the interquartiles range.

Therefore, adding a picoinjection module to the workflow enabled a fivefold gain in gene detection rates compared to inDrop (4,926 vs 1,016, Fig. 3A), with increased RNA biotype diversity and intron detection. The sensitivity is conserved throughout different input formats, which makes spinDrop a versatile and sensitive platform for scRNA-seq.

## Profiling of the embryonic mouse brain using a highly damaged sample input

We further investigated whether the sorting feature of the spinDrop platform could be leveraged to improve the transcriptomic quality of biological samples with low viability. Such samples contain a larger amount of degraded or disintegrated cells, or cells in stressed states that might not reflect a physiologically relevant transcriptional programme[21,43]. To this end, we dissected and dissociated the brain of mouse embryos at developmental stage E10.5 and left it at room temperature for 3 hours to decrease viability (Fig. 4A). This treatment led to 36.6% viable cells after Calcein-AM staining and counting on a haemocytometer. We then processed these cells with and without sorting to demonstrate the utility of our workflow to tackle complex samples. Ranked read coverages per barcode (knee plots) were generated for both samples using zUMIs[44], which underlined a clear inflexion point for the sample where pre-sorting was used, which contrasted strongly with the unsorted sample which had a linear trend (Fig. 4B). This rebalancing towards bona fide cells with increased coverage suggests an enrichment for high-quality material using FADS sorting.

The datasets were then further investigated using DropletQC[19] to identify empty droplet barcodes as well as dead cells, computed using nuclear fractions (ratio of unspliced to spliced reads) on both sorted and unsorted samples. The analysis revealed that 90.7% and 1.2% of the barcodes were determined as empty droplets and damaged cells in the unsorted population, respectively (Supplementary Fig. 4H). By contrast, these proportions were 4.6% and 40.2% with FADS sorting (Supplementary Fig. 4I), illustrating efficient elimination of spurious transcripts by discarding empty droplets using FADS (Fig. 4C). To verify if the datasets generated would be amenable to transcriptional atlasing of heterogeneous cell types, the spinDrop datasets were integrated with mouse embryo references generated using the 10x Chromium v1[45] (E11) and sci-RNA-seq3[11] (E10.5, downsampled to 20,000 cells to match other sample sizes) methods as references. To this end, Seurat v3 was utilised to compute integration anchors and generate a shared embedding[46] between all datasets, and further computed Pearson correlation coefficients between the clusters in the shared embedding and the annotation from La Manno et al.[45] were used to transfer labels to all remaining datasets (see Methods) (Fig. 4D, Supplementary Fig. 4J). The cell type distributions after label transfer delineated a decrease in the proportion of cells in the low complexity cluster ("Bad cells" in yellow in Fig. 4D) when applying FADS sorting to the input population, from 50.9% without sorting to 21.7% with sorting and 26.1% with the 10x Chromium dataset (Fig. 4E). Sorting using FADS did not affect cell-type representation (apart from a slight overrepresentation of fibroblasts in the FADS dataset) or the proportion of cells in each cell-cycle phase, showing that the method is broadly applicable to cell atlasing and does not affect cell-type representation (Supplementary Fig. 4E, G).

To further investigate if the dataset generated using our methodology could improve marker delineation in the dataset, a Wilcoxon rank sum test was used to compute differential marker analysis between the 10x and spinDrop datasets (with sorting) on the neuroblast population (Supplementary Data 3). The analysis revealed an enrichment for core cortical maturation markers, such as *Sox11*[47], *Sox2ot*[48] and *Neurod1*[49] using the spinDrop methodology. Out of the top 100 markers overexpressed in the neuroblasts profiled using spinDrop, 43 were part of the "neural system development" gene ontology term (GO:0007399, FDR = 3.1e⁻¹⁵), underlining core cell-type specific mechanism that were absent in the equivalent 10x Chromium dataset.

These findings underline the capabilities of our workflow to capture the spectrum of heterogeneity in a damaged input sample by removing damaged cells and empty droplets from the sampled population.

## High-throughput nascent RNA sequencing using spinDrop

SpinDrop is a multi-step method enabling complex RNA capture protocols that may not be attainable using single-step standard droplet tools because of temperature, enzymatic and buffer incompatibilities[12,14,15]. For example, performing reverse cross-linking in droplets is currently prohibited as mentioned previously. This bottleneck prevents the scaling-up of methods that use fixation as a means to broaden transcriptome read-outs, such as metabolic labelling methods like scEU-seq[50]. We therefore applied the methodology to uncover nascent RNA transcription during mouse organogenesis at high-throughput (referred to as 5EU-seq in the remainder of the text). To achieve this, mouse embryos at E8.5 were dissected and cultured in vitro for 3 h with the 5EU (5-Ethynyl Uridine) analogue, then dissociated, fixed and subjected to click chemistry to add biotin groups to the analogue incorporated in the nascent transcript. The fixed cells were then processed using spinDrop for reverse cross-linking and reverse transcription, without sorting, and the nascent transcripts were pulled-down using streptavidin-coated beads before performing library preparation on both nascent and non-nascent transcripts (Fig. 5A). Nascent and non-nascent RNA molecules for each cell were then linked bioinformatically. The nascent RNA libraries displayed a higher proportion of UMIs mapping to intronic regions of transcripts than the non-nascent library (43.4% versus 21.4%, ratios of median UMI counts), confirming higher capture rates for nascent transcripts (Fig. 5B, Supplementary Fig. 5B, C). The proportion of nascent to non-nascent UMIs was 18%, which echoes the proportion of unspliced to spliced UMIs identified using an equivalent E8.5 inDrop dataset[51] (15%) (Supplementary Fig. 5A).

Next, cell-type annotations from a sci-RNA-seq3 dataset at E8.5 were projected on a shared embedding with the spinDrop dataset as described in the previous section (Fig. 5C). Because the spinDrop

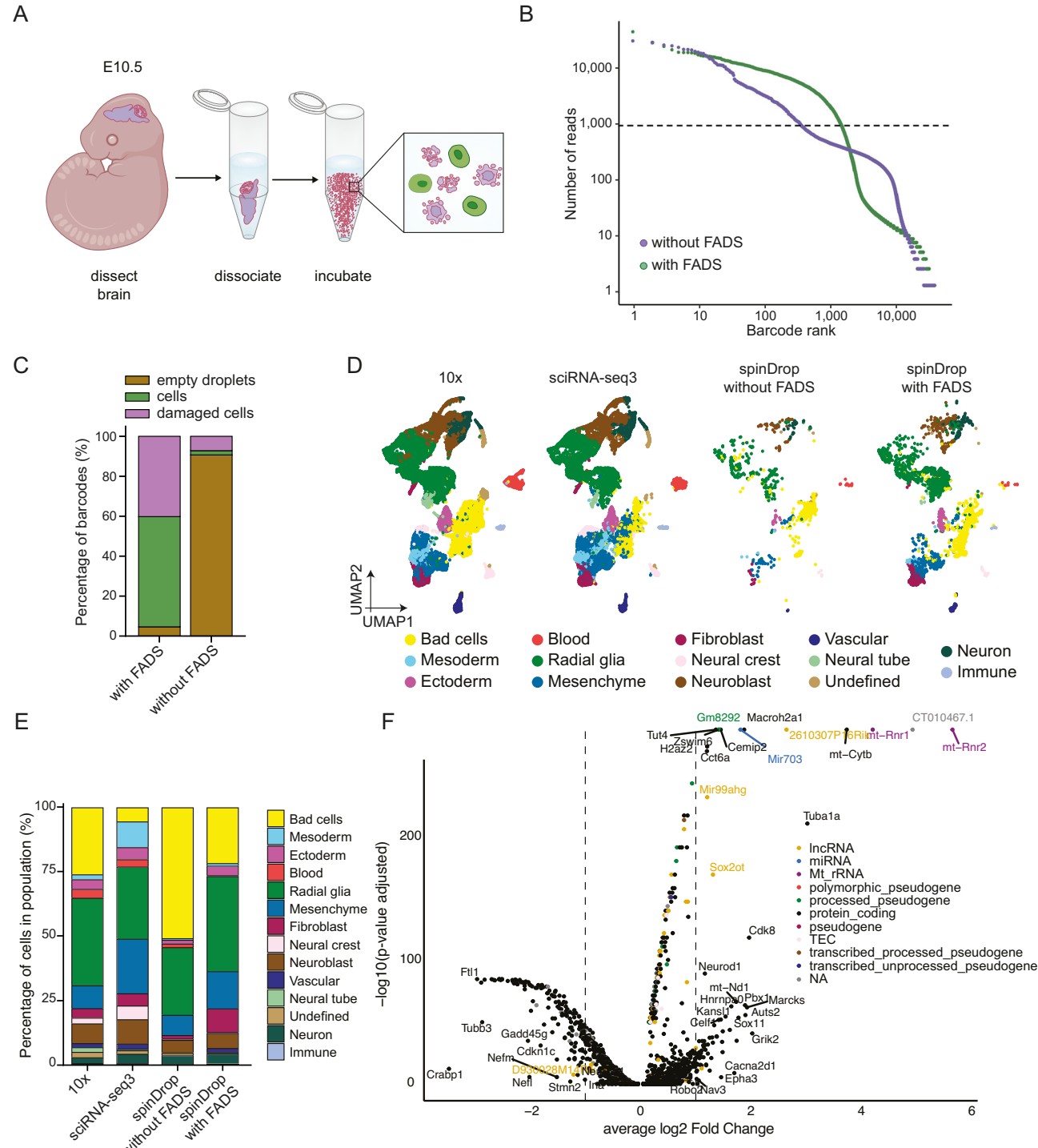

**Fig. 4 | Transcriptional atlas of embryonic mouse brains at E10.5 generated using the spinDrop methodology. A** Schematic of the cell recovery process and staining. The cells are stained using Calcein-AM for separating live cells (green) from dead cells (purple). The cells were left in PBS at room temperature for three hours after dissociation to increase cell death rates. **B** Barcode rank knee plot for the mouse brain dataset using in-line sorting to extract live cells (green, with FADS), or without sorting to denote results using standard droplet microfluidic scRNA-seq (purple, without FADS). The sorted population shows a barcode inflection point at ~1000 barcodes, contrary to the sorted population, illustrating lower background from dead cells and cell-free RNA. **C** Results from barcode quality control inspection using the DropletQC tool for both the droplets with and without FADS sorting.

The sorted population shows a significantly lower proportion of empty droplets and damaged cells compared to the unsorted population. **D** UMAP dimensional reduction on an equivalent 10x Chromium, sci-RNA-seq3 and spinDrop dataset (with and without droplet sorting), with cells labelled by major cell type. $n = 30,244$ cells for the 10x dataset, 20,000 cells for the sci-RNA-seq3 dataset, 2472 cells for spinDrop with FADS sorting and 802 cells for spinDrop without FADS sorting. **E** Histogram representing the proportion of cell types for each technology. **F** Volcano plot representing the differentially expressed genes for the neuroblast cluster between the 10x Chromium (negative log₂ fold change values) and spinDrop with sorting (positive log₂ fold change values) methods. Source data are provided as a Source Data file.

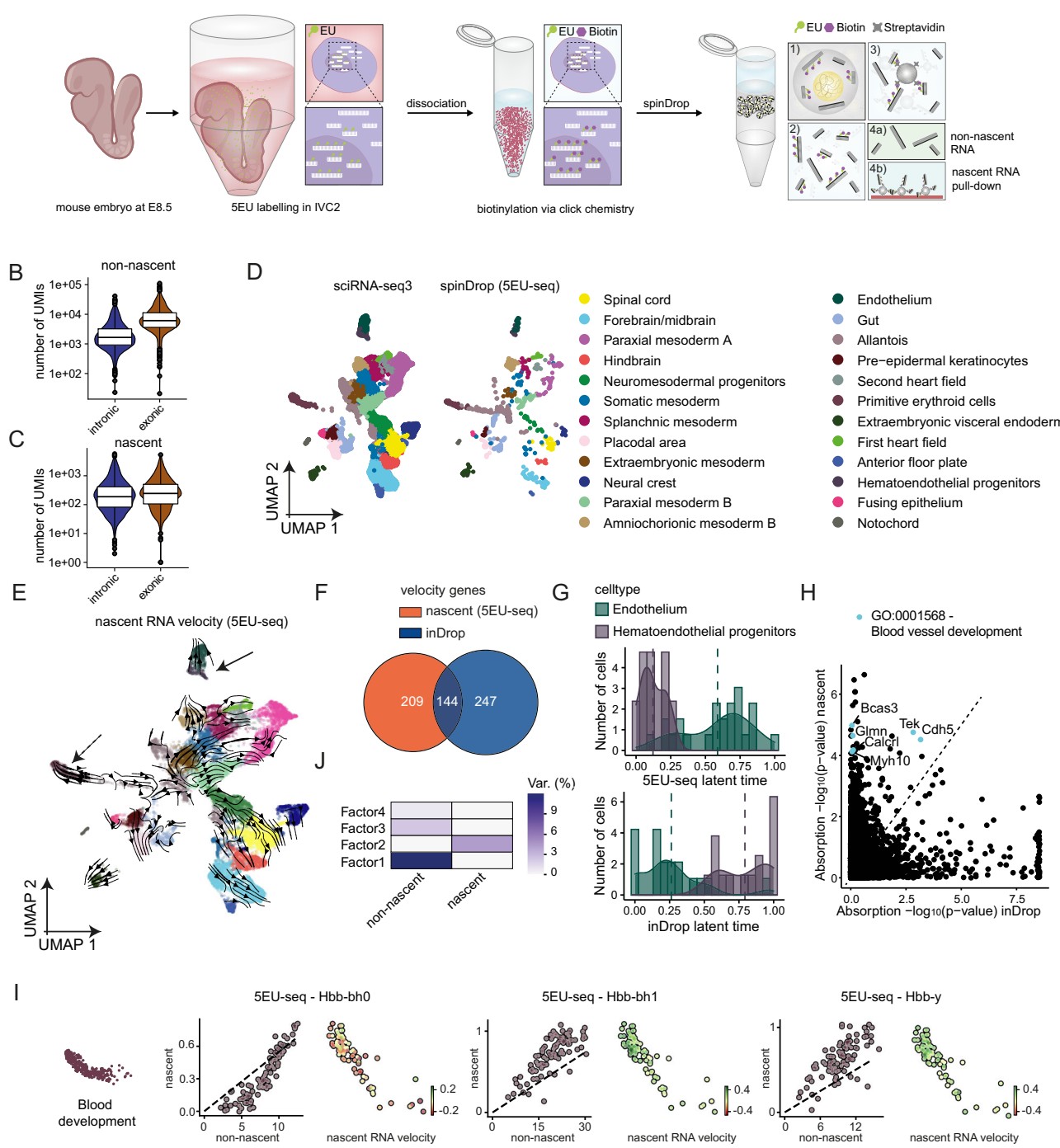

dataset was filtered to contain barcodes represented both in the nascent and non-nascent libraries, proportions of cell types compared to the sci-RNA-seq3 reference may inform on the capabilities of the analogue to diffuse throughout the embryo. For example, primitive erythroids, gut, endothelium and extra-embryonic mesoderm and endoderm cells were proportionally enriched in the 5EU-seq dataset, whereas neural cell types such as the spinal cord, forebrain/midbrain and hindbrain were depleted (Supplementary Fig. 5D). The fraction of nascent RNA reads was also smaller in tissues of neural origin, and higher in cell-types from mesodermal origin, in particular the somatic mesoderm, which displayed a high fraction of nascent RNA reads (24%) underlying tissue-specific capture of transcriptional dynamics (Supplementary Fig. 5D). Then, velocity vectors were computed via scVelo[39]

using the nascent and non-nascent transcripts as an input for the spinDrop dataset and unspliced and spliced transcripts for the equivalent inDrop dataset and were projected on a shared UMAP embedding (Fig. 5E and Supplementary Fig. 5E), yielding a list of dynamical genes with 144 intersecting and 456 non-intersecting genes (Fig. 5F and Supplementary Data 4). Major trajectories, such as the bi-potent commitment of neuromesodermal progenitors to spinal cord and paraxial mesoderm or the differentiation of haematoendothelial progenitors to endothelium could be observed with nascent RNA capture, whereas these trajectories were less clearly defined in the inDrop dataset (Supplementary Fig. 5E). To quantify this finding, latent time values across the endothelial development trajectory were computed and accurately retraced natural development by computing

**Fig. 5 | Nascent RNA sequencing using 5EU-seq defines transcription dynamics during mouse organogenesis. A** Schematic of the embryo staged at E8.5 processing using the droplet-based 5EU-seq method. The CD-1 embryos are incubated for 3 hours in IVC1 in presence of the 5EU analogue. The cells are then dissociated, fixed, permeabilized and a biotin group is added using click chemistry. After processing with spinDrop, the recovered cDNA molecules are incubated with streptavidin-coated magnetic beads to separate nascent and non-nascent RNA/cDNA complexes. **B** Number of UMIs detected for the non-nascent fraction mapping to either intronic or exonic regions. Data in the box plot represent the 25%, median (centre) and 75% percentiles with minimum and maximum values. **C** Number of UMIs detected for the nascent fraction mapping to either intronic or exonic regions. Data in the box plot represent the 25%, median (centre) and 75% percentiles with minimum and maximum values. **D** Integration of the spinDrop dataset with the sc-iRNA-seq3 mouse embryo dataset (downsampled to 20,000 cells) with the 5EU-seq data for label transfer shows capture of all main cell types. **E** Single-cell nascent RNA velocity field computed using scVelo, projected on the shared embedding (spinDrop and sci-RNA-seq3). The full arrow indicates

endothelium maturation, the dashed arrow indicates blood maturation. **F** Number of intersecting dynamical genes detected using scVelo in stochastic mode using unspliced and spliced matrices for the downsampled inDrop dataset ($n = 936$ cells) and using the nascent and non-nascent RNA matrices with the spinDrop dataset. **G** Latent time projections along the endothelial maturation trajectory, from haematoendothelial progenitors to endothelium, were calculated using RNA velocity in the inDrop dataset or nascent RNA velocity in the spinDrop dataset. **H** Lineage driver gene absorption probabilities for the endothelial maturation trajectory (-$\log_{10}$ transformed) calculated using CellRank for the inDrop (RNA velocity) and spinDrop (nascent RNA velocity) datasets, highlighting an enrichment ($p$-value < $10^{-5}$) for gene ontology terms relating to blood vessel development in the nascent RNA dataset. **I** Nascent RNA sequencing uncovers haemoglobin transcriptional kinetics. From left to right, phase plots and velocity values superimposed on the UMAP projection for blood progenitors of *Hbb-bh0*, *Hbb-bh1* and *Hbb-y*. **J** Multi-omic factorial analysis (MOFA) of nascent and non-nascent RNA matrices underline nascent RNA-specific variance in Factor 2. Source data are provided as a Source Data file. Var. strands for variance.

velocity values using nascent RNA, whereas the trajectory was inverted in the inDrop dataset (Fig. 5G) using conventional velocity measurements. In addition, the absorption probabilities computed using CellRank[52] showed an enrichment for lineage drivers relating to blood vessel development (GO:0001568, FDR = 3*10⁻²) using nascent RNA sequencing (Fig. 5H, Supplementary Data 5), showcasing accurate delineation of developmental drivers. For some genes, RNA velocity did not capture expression dynamics altogether due to the lack of measurable unspliced molecules in the inDrop dataset. This can be explained by the stochastic nature of barcoded oligo(dT) probe binding to A/T-rich unspliced sequences across the transcript body[38], hereby introducing a sequence-bias for velocity computation. In addition, the enrichment for 3' molecules favours 3'-proximal intronic regions, which is highly gene-specific. Nascent RNA sequencing, on the other hand, performed better for the set of genes highlighted in Fig. 5F because it does not rely on stochastic probe binding and bears no sequence bias, meaning that nascent transcripts are being read at the 3' end of the transcripts similarly to non-nascent transcripts. One example is the morphogen *Shh* which was activated in the placodal area and notochord (Supplementary Fig. 5F), whereas RNA velocity could not be inferred in the inDrop dataset. Similarly, haemoglobin synthesis dynamics could exclusively be captured using nascent RNA sequencing, showing a repression of *Hbb-bh0* and induction of *Hbb-bh1* and *Hbb-y* in blood progenitors at E8.5 (Fig. 5J). Some gene dynamics were detected both using nascent RNA sequencing and RNA velocity, but dynamics were not representative of natural development using the latter method, such as the erroneous split induction and repression of *Pax6* in the spinal cord using RNA velocity which was accurately identified as being solely inducted using nascent RNA measurements (Supplementary Fig. 5I). To further determine the unique contributions of nascent RNA towards each cell's transcriptome, multi-omic factorial analysis (MOFA)[53] was performed on both normalised nascent and non-nascent matrices, which delineated clear contributions of the nascent RNA fraction towards factor 2 variance. Further inspection of the weights contributing to factor 2 highlighted key contributions of transcription factors *Gata2* and *Tfap2c*, underlining core morphogenetic signatures captured via nascent RNA sequencing (Fig. 5J and Supplementary Fig. 5H, I).

These findings position spinDrop as a modular methodology that may provide more comprehensive '-omic' maps of heterogeneous tissues than current state-of-the-art methods.

## Discussion

spinDrop is an open-source droplet microfluidic workflow that uses droplet sorting and picoinjection to maximise bona fide information output from high-throughput single-cell experiments. We demonstrate that spinDrop enables deterministic sequencing of viable single-

cells from the pool of empty droplets or droplets containing cell debris and damaged cells, which in turn reduces sequencing cost and removes artefacts from single-cell experiments. The method also supports cell type enrichment using fluorescently-labelled antibodies targeting cell surface markers, which may prove useful on samples where FACS cannot be employed due to cell number or viability concerns. In addition, gene detection rates were increased fivefold compared to inDrop[14] to match the sensitivity of the leading commercial platform (10x Chromium). The cost bottleneck of high-throughput single-cell RNA-seq was addressed in spinDrop, with a 20 to 50% reduction in sequencing cost depending on input sample quality, and 6.2 decrease in library preparation cost compared to the 10x Chromium, leading to an average 60% decrease in overall cost per cell (0.4 USD per cell for spinDrop, 1 USD per cell for the 10x Chromium, Supplementary Data 6)[12]. In addition, functionality across different input modalities was demonstrated for HEK293T cells, with high performance being achieved for whole cells, extracted nuclei and fixed cells. This will enable single-cell processing of archival tissues, which is a critical need in the clinic[42,54]. Although fixed cell processing has been demonstrated using Drop-seq[55] and the 10x Chromium[56], these applications suffer respectively from low cell capture (10-fold lower than spinDrop) and reliance on probe-based capture which is species-specific and prevents genotyping applications which is crucial for investigating clinical samples. spinDrop capabilities were further demonstrated by profiling a damaged sample of the developing mouse brain at E10.5, removing the noise arising from empty droplets and damaged cells, thus showcasing the suitability of the method for molecular atlasing of complex and low input samples (Supplementary Data 7). The number of cells analysed in this study (9,599) with spinDrop is in line with previous droplet proof-of-concepts (10,000 cells for inDrop[14], 945 cells for DisCo-seq[57] and 498 cells for scRNA-seq combined with printed droplet microfluidics[58]). 10x Chromium has since then released a v3 formulation that increases median gene capture rates by 29.9% at maximum saturation on HEK293T cells according to their application note; further widening the gap between in-house and commercial scRNA-seq implementations. Therefore, further optimisations, such as utilising dissolvable beads for barcode release and fine-tuning barcode concentration[32], could be implemented in the spinDrop workflow to further boost efficiency to match the latest 10x Chromium v3 improvements.

Application of quality control filtering methods to bioinformatically remove noise from single-cell datasets is standard procedure before performing downstream analysis[59]. However, several drawbacks are associated with computational filtering of noisy cells from datasets. For example, damaged cells and cell debris can be filtered out using gene expression counts mapping to mitochondrial RNA molecules, as they are retained at higher rates than cytoplasmic RNA when

the cell membrane is perforated. However, mitochondrial RNA content varies with cell type and species[60], which means that filtering on pre-set thresholds may alter the dataset and bias subsequent interpretation. Another metric to identify empty droplets and damaged cells in a dataset is generated by computing the nuclear fraction ratio of unspliced-to-spliced reads[19]. Because empty droplets contain higher levels of spliced cytoplasmic RNA and damaged cells contain higher levels of unspliced nuclear RNA, the latter can be filtered out by setting thresholds on nuclear fraction (ratio of unspliced to spliced UMI counts). However, nuclear fractions may, again, differ widely between the profiled cell types. For example, erythrocytes contain small traces of mature RNA molecules and may therefore be identified as empty droplets using current tools[19]. Fluorescence-activated sorting of droplets containing viable cells at the moment of encapsulation circumvents this issue and deterministically resolves viable cells, independently of mitochondrial RNA content or cell type. Another crucial pre-processing step consists in the removal of cellular doublets from the dataset. However, current tools have variable performance depending on cell-type and mostly identify heterotypic doublets[24,25] (i.e. multiplets from different cell types). Here, we demonstrate a proof-of-principle approach for discarding doublets using FADS by applying an upper threshold on fluorescence. Sorting metrics showed reduced co-encapsulation rates compared to predictions, illustrating the potential of the method to remove doublets from the sequenced pool. Deterministic sequencing of droplets containing single-viable cells or intact nuclei therefore reduces artefacts introduced by downstream processing, but also alleviates the sequencing cost associated with these artefacts. However, the approach does not remove ambient RNA from sorted droplets, hence methods like SoupX[17] may still be required to remove ambient RNA co-encapsulated with cells from the dataset. Reduction of droplet size in the future (i.e. high cell volume to droplet volume ratio) may provide a viable route to remove or dampen the effects of such artefacts in future implementations.

Empty droplet removal in the spinDrop workflow will further enable counting of synthetic molecular RNA spike-ins (such as ERCC[41] or molecular spikes[61]) along single-cell gene expression measurements. Indeed, current workflows are constrained by Poissonian loading of single-cells into droplets; forcing high capture of synthetic RNA species in empty droplets which would increase sequencing costs significantly, largely explaining why this gold standard approach has not been transferred from plate-based to droplet-based assays. Removing empty droplets from the analysis circumvents these issues, and offers new avenues for removing counting biases from single-cell matrices in high-throughput datasets, drastically increasing the accuracy of any single-cell experimentation.

Although some methodologies have utilised droplet sorting for single-cell sequencing applications, they are limited in their applicability to molecular atlasing. Studies by Clark et al.[62] and Zhang et al.[58] are primarily utilising sorting for cell-type specific isolation rather than the enrichment of single viable cells[16]. Machine vision methods have been described for the deterministic sorting of co-encapsulated single-cells and single barcoded microgels, but current workflows suffer from low throughput (77-fold smaller than spinDrop)[57] and do not take cell viability into account during sorting[57]. Furthermore, none of the aforementioned methodologies have been designed to increase sensitivity compared to other methods to match the leading commercial standard (10x Chromium), and the protocols have not been demonstrated across input modalities for the processing of archival tissues and nuclei, thus limiting their applicability towards molecular atlasing endeavours across sample types. In addition, the advantages of excluding damaged cells and empty droplets were not explored using these methodologies. Although sorting did not significantly skew cell-type proportions in our profiling endeavours, future implementations of the sorting technology could comprise an image analysis component to more faithfully discard cell doublets from large cells, or even

add a phenomic component to the sequencing results, e.g. based on cell size or morphology. Although spinDrop requires a picoinjection step after cell sorting, the throughput of this microfluidic handling step is manageable (up to 70 Hz, equivalent to 252,000 cells/hour). Furthermore, at this stage empty droplets had already been discarded during sorting. As a consequence the picoinjection operation is exclusively performed on droplets containing cell lysates, which means that the demands in terms of throughput for this step are substantially lower than for the initial droplet formation.

More generally, we anticipate that the powerful combination of droplet sorting and picoinjection described here will complement microfluidic methods that go beyond transcriptome sequencing in droplets, such as ATAC-seq[32] or multi-omic profiling[63]. We demonstrate the benefit of multi-step microfluidics by performing reverse-crosslinking in droplets, which allowed us to profile nascent RNA transcription during mouse organogenesis using 5EU-seq, uncovering previously hidden layers of biology that are not attainable using state-of-the-art methods. Similarly, increased mitochondrial transcriptome coverage in spinDrop may benefit lineage tracing endeavours[64]. Integration with commercial toolboxes like the 10x Chromium will present an opportunity to decrease cost and increase customer acquisition further. Future droplet sorting implementations may include multi-modal sorting, such as fluorescence coupled to image-based sorting[65] to further decrease doublet rates and acquire cell-type phenomic profiling, which would enable higher cell loading concentrations and increased throughputs as well as multimodal phenotypic characterisation. Image-based sorting in addition to fluorescence would also help validate if the method depletes specific populations. In our mouse brain analysis and nascent RNA analyses, neural crest cells were slightly depleted using spinDrop (2.9%) compared to sci-RNA-seq3 (3.8%). Although neural crests have been shown to be enriched in nuclei data compared to whole-cell using a neuroblastoma clinical sample[66], further validation of selected droplets via imaging of cells being sorted would be beneficial to cell atlasing efforts. Sorting throughputs might be increased by using smaller beads that would provide better droplet monodispersity by reducing the droplet volume, or using serial electrodes that can improve sorting speed[67]. Further implementations may also include the use of dissolvable hydrogels[32] or enzymatically released barcodes[68] from the solid-support microgels to circumvent potential limitations of the current system, which uses UV-induced barcode release which may reduce RNA capture due to cross-linking.

Overall, spinDrop is well aligned with atlasing efforts like the Human Cell Atlas, as it provides 'ground truth' and sensitive reference datasets of human biology. In addition, the method unifies low-cost, high-throughput and high-sensitivity, which is critical to the success of the community for accurately determining the spectrum of heterogeneity in a sample.

# Methods

## Ethical Statement

All experiments performed were under the regulation of the Animals (Scientific Procedures) Act 1986 Amendment regulations 2012 and were reviewed by the University of Cambridge Animal Welfare and Ethical review body (AWERB). Experiments were also approved by the Home Office.

## Design of the droplet generation device

The integrated device for compressible barcoded bead and single-cell co-encapsulation into droplets, followed by fluorescence-activated droplet sorter (FADS,Supplementary Fig. 1A) incorporates a modified droplet generator architecture used previously[14,28,37] with an added FADS module for enrichment of droplets containing single viable cells stained with Calcein-AM, Vybrant Green or fluorescently-labelled antibodies. The FADS module is based on integrated fibres[69] for both excitation and

detection of fluorescence. The emission light is collected by the detection optical fibre and transferred to the detector tube housing a set of emission filters mounted in front of the detector of the photomultiplier tube (Supplementary Fig. 2A). When a fluorescent signal exceeds an arbitrarily set threshold, a high voltage pulse is generated (1 kV) and delivered to the microfluidic sorting junction via 'salt electrodes' filled with a 5 M NaCl solution. As a result, highly fluorescent droplets with live cells are derailed to the positive 'hit' collection channel. The cell encapsulation and sorting chip comprises sections of increasing depths: i) the droplet generator is 75 μm deep to ensure good encapsulation efficiency of 60 μm barcoded beads, ii) the depth of the detection spot (95 μm) is determined by the width of the fibres, and iii) the sorting junction and collection channles are deeper (175 μm) to avoid squeezing the droplets and to facilitate their redirection to the positive channel. Additionally, a gapped divider was implemented at the sorting junction[70] to gradually push droplets to the outlet channels and minimise the risk of droplet rupture. The microfluidic device additionally comprises two inlets (number 6 and 7 Supplementary Fig. 1A) and a flow-focusing junction for the generation of 'buffer droplets' that do not contain cells, beads or ambient RNA. The role of buffer droplets is to facilitate handling of the collected material before subsequent picoinjection. The picoinjection module is an improved version of a device used in a previous study[37]. However, the inlet for the emulsion diluting oil (number 2 Supplementary Fig. 1B) is now located behind the droplet emulsion inlet (number 1 Supplementary Fig. 1B) which reduces the fragmentation of densely packed droplets before reinjection. The final section of the picoinjection device was deepened (180 μm) to stabilise droplets and reduce their merging during the abrupt change in depth between the shallow microfluidic channel and the wide collection tubing.

## Photolithography of microfluidic moulds

The channel layout for the microfluidic chips was designed using AutoCAD (Autodesk) and printed out on a high-resolution film photomask (Micro Lithography Services). The designs in Supplementary Fig. 1 can be accessed at https://openwetware.org/wiki/DropBase: Devices and can be found in the Supplementary Software 1 file. The microfluidic devices were fabricated following standard photo- and soft lithography protocols that can be performed in cleanrooms or outsourced to contract manufacturing companies. First, microfluidic moulds were patterned on 3" silicon wafers (Microchemicals) using high-resolution film masks (Micro Lithography Services) and SU-8 2015, 2075 and 2100 photoresists (Kayaku Advanced Materials). A MJB4 mask aligner (SÜSS MicroTec) was used to UV expose all the SU-8 spin-coated wafers. The thickness of the structures (corresponding to channel depths in the final microfluidic devices) was measured using a DektakXT Stylus profilometer (Bruker). The settings used for photolithography can be found in Supplementary Data 8 and 9, for the FADS sorter and picoinjector, respectively.

## Soft lithography PDMS chip fabrication

To manufacture PDMS microfluidic devices, 20-30 grams of silicone elastomer base and curing agent (Sylgard 184 Dow Corning) were mixed at a 10:1 (w/w) ratio in a plastic cup and degassed in a vacuum chamber for 30 minutes. PDMS was then poured onto a master wafer with SU-8 structures and cured in the oven at 65 °C for at least 4 hours. Next, the inlet holes were punched using two types of biopsy punches with plungers (Kai Medical): a 1.5 mm diameter punch was used to make the inlet for cell delivery tip (number 2 Supplementary Fig. 1A), outlet tip for droplet collection (number 9 Supplementary Fig. 1A) and the inlets for droplet reinjection (number 1 Supplementary Fig. 1B), while other inlets were inserted using a 1-mm-wide biopsy puncher. The FADS PDMS chip was first plasma bonded to an approximately

1-mm thin PDMS slab (cured beforehand) and then to a 52 mm x 76 mm x 1 mm (length x width x thickness) glass slide (VWR) in a low-pressure oxygen plasma generator (Femto, Diener Electronics). As a result, we obtained a 3-layer device with a patterned PDMS on top, a thin PDMS slab in the middle, and a glass slide at the bottom. The picoinjection chip was bonded only to the glass slideresulting in a two layer device. Next, the hydrophobic modification of the microfluidic channels was performed by flushing both types of devices with 1% (v/v) tri-chloro(1H,1H,2H,2H-perfluorooctyl)silane (Sigma-Aldrich) in HFE-7500 (3M) and baked on a hot plate at 75 °C for at least 30 minutes to evaporate the fluorocarbon oil and silane mix. Next, the fabricated FADS device was mounted on the microscope stage for the assembly of fibres and tubing and the picoinjection chip was used in the second stage of the microfluidic experiment.

## Detailed step-by-step protocol for the microfluidic rig and the FADS chip assembly

The setup is a modification of a previously presented system for fluorescence-activated droplet sorting[71]. Here we use fibres for the excitation and detection of fluorescence from droplets on a chip. The integration of fibres with the chip and the hardware is described in detail below. Setup procedure of the instruments for the droplet sorter:

1. An inverted microscope (Olympus IX73) and a 488 nm laser (25 mW, Stradus Vortran) were first installed on a breadboard plate (MB6090/M, Thorlabs) placed on a laboratory bench.
2. A fast camera (Miro eX4, Phantom) was attached to the camera port of the microscope using a C-mount adapter (SM1A10, Thorlabs) and an SM1 lens tube (SM1L20, Thorlabs) with an N-BK7 Plano-Convex Lens, Ø1", f = 50mm (LA1131-A, Thorlabs) (L3 Supplementary Fig. 2A) that was placed to the tube 50 mm before the detector of the camera.
3. A 593 nm long pass filter F1 (FF01-593/LP-25, Semrock) was inserted into the microscope condenser (L1 Supplementary Fig. 2A) to allow for red light illumination of the microfluidic device with the microscope lamp. During the experiments, the transmitted red light was collected by the objective (usually 10x or 20x), leaving the microscope through the camera port to be recorded by a fast camera.
4. A neutral density filter OD = 0.5 (NE05A, Thorlabs) was installed in front of the 488 m laser to reduce the power laser beam (ND, Supplementary Fig. 2A). The filter was attached to the 30 mm cage plate (CP33/M, Thorlabs) that was mounted on a breadboard using Ø12.7 mm optical post (TR20/M, Thorlabs), pedestal post holder (PH20E/M, Thorlabs) and a clamping fork (CF125, Thorlabs).
5. A module for coupling the light into the optical fibre (C1 Supplementary Fig. 2A) was assembled in line with the laser beam. The module was built using the following Thorlabs components: (i) adjustable FC/PC collimator (CFC-11X-A), (ii) SM1-threaded adapter (AD9.5 F), (iii) XY translating mount for Ø1" optics with a quick release plate (CXY1QA), (iv) SM1 coupler with external threads (SM1T2), and (v) kinematic mount for Ø1" optics (KM100T). The module was mounted on a breadboard using a Ø12.7 mm optical post (TR20/M, Thorlabs), a pedestal post holder (PH20E/M, Thorlabs), and a clamping fork (CF125, Thorlabs). The module was aligned with the laser beam by attaching the fibre (M43L02 Thorlabs) to the FC/PC collimator and changing the tilt angles of the kinematic mount and the XY position in the translating mount until the output power of the light exiting the fibre was at least 90% of the input power after passing ND filter. The power of the laser used for alignment was set to below 1 mW for safety reasons, and the USB power meter (PM16-130, Thorlabs) was used to measure intensity of light beam emering from the fiber.

6. A PMT detector module was next assembled in the proximity of the microscope (C2 Supplementary Fig. 2). The module was built of the following Thorlabs components (counting from the fibre connector): i) adjustable FC/PC collimator (CFC-11X-A), ii) SM1-threaded adapter (AD9.5 F), iii) tube housing the set of filters (F2 Supplementary Fig. 2a) composed of 1-notch (NF488-15, Thorlabs) and two bandpass filters (FF01-550/88-25, Semrock), and iv) N-BK7 Plano-Convex Lens, Ø1", f = 50 mm (LA1131-A, Thorlabs) (L2 Supplementary Fig. 2a) mounted in SM1 lens tube (SM1L20, Thorlabs) 50 mm before the detector of photomultiplier tube PMT (PM002, Thorlabs). The whole module was mounted on a breadboard using a Ø12.7 mm optical post (TR50/M, Thorlabs), a pedestal post holder (PH40E/M, Thorlabs), and a clamping fork (CF125, Thorlabs).

   For sorting of cells labelled with IgM-PE antibodies a different, 561 nm laser (50 mW, OBIS, Coherent) and other filters were used: F1 – 635 nm longpass filter (BLP01-635R-25, Semrock), F2 - a set composed of one longpass filter 561 nm (BLP02-561R-25, Semrock) and two bandpass 593/40 filters (FF01-593/40-25, Semrock).

7. Pulse generator PG (TGP110, Thurlby Thandar Instruments), function generator FG (TG2000, Thurlby Thandar Instruments), and high-voltage amplifier AMP (610E, Trek) were placed on the shelf above the microscope (PG, FG, AMP in Supplementary Fig. 2a).

8. BNC cables (2249-C-24, Thorlabs) were used to connect the output of the pulse generator to the input of the function generator, which output was next connected to the high-voltage amplifier.

9. The blunt end of the high voltage cable was soldered to the BNC connector (546-4875, RS), which was next connected to the adapter with a female BNC to test clips connector (T3788, Thorlabs).

10. A PCIe-7841R FPGA device (National Instruments) installed previously in the desktop computer was connected to SCB-68A connector block (National Instruments) via SHC68-68-RMIO Shielded Cable (National Instruments). Two analogue output (AO) pins (gain voltage and ground) of SCB-68A were connected to the PMT detector to control its sensitivity. The PMT device was also wired to the analogue input (AI) pins to transfer the raw voltage signals from measured fluorescence.

11. Two other digital input/output (DIO) pins were connected to the BNC cable that was then split to simultaneously trigger the pulse generator and the fast camera.

    Preparation of optical fibres and their integration within the microfluidic chip for encapsulation and sorting:

12. Two different optical fibres were used for the assembly of the chip: i) the excitation light fibre (M94L02, Thorlabs) with a cladding diameter of 125 μm and a core diameter of 105 μm with a numerical aperture (NA) of 0.1 and ii) the detection fibre (M43L02, Thorlabs) with a cladding diameter of 125 μm and a core diameter of 105 μm with NA of 0.22.

13. Both fibres were cut at their ends to remove one of the FC/PC connectors, and next, the outer protective PVC jacket was removed using a three-hole fibre stripper (FTS4, Thorlabs). The Kevlar protective threads were cut with a scalpel and finally, acrylate coating was removed using a fibre stripping tool (T06S13, Thorlabs).

14. Then, the FC/PC end of the detection fibre was connected to the module for coupling the light into the optical fibre. The tip of the fibre tip was cleaved using a ceramic fibre scribe (CSW12-5, Thorlabs) in order to obtain a flat tip end. The quality of the cleavage was inspected by passing a low-power (e.g. 0.1 mW) laser light through the fibre and visual inspection of the shape of the beam emerging from the fibre tip end. If necessary, the cleavage was repeated until a spherical light beam shape was observed. The fibre was then connected to the FC/PC collimator in the PMT detector module.

15. Next, the excitation fibre was connected to the module for coupling the light to the optical fibre and cleaved similarly to the detection fibre.

16. Next, both optical fibres were inserted to the sorting chip as presented in Supplementary Fig. 1D, photographs a)-c). Fixing of fibres to the chip was performed on the microscope stage, and the microscope's camera was used to verify the position of the fibre ends. First, the microfluidic channels housing the fibres were filled with HFE-7500 oil (3M), and then fibre tips were manually inserted into the chip. Fibres were stabilised by attaching them to the glass slide with pieces of insulation tape.

17. Before the experiment, the input intensity of the 488 nm laser was set to 10 mW, which translates to approximately 3 mW on the chip after passing the ND filter. For sorting of cells labelled with IgM-PE antibodies the input intensity of the 561 nm laser was set to 25 mW, resulting in approximately 8 mW on the chip, after passing the ND filter and coupling to the fibre.

18. In the final step, syringes containing 5 M NaCl were connected via polyethylene tubing to the device (Supplementary Fig. 1D, photograph d) and the electrode channels pre-filled with salt solution filtered as previously described[70]. The test clips adapter from the high-voltage cable were connected to the steel needles of the plastic 5-ml syringes (BD) with a salt solution.

Alternative commercial instrumentation may also be employed to achieve FADS and picoinjection using push-button solutions, such as the Styx (FADS) and Onyx (picoinjection) from Atrandi Biosciences. A cost breakdown of the main components for building the spinDrop rig are given in Supplementary Data 10.

### mESC and HEK293T cell culture and preparation

HEK293Ts (gifted from Dr. Marc de la Roche, Department of Biochemistry, University of Cambridge) were passaged every second day and cultured in T75 flasks. The culture media was DMEM (4500 mg/L gluc & L-glut & Na bicarb, w/o Na pyr, Sigma-Aldrich) supplemented with 10% heat-inactivated FBS and 1x Penicillin-Streptomycin (Sigma-Aldrich). For passaging and collection, the cells were washed with 10 mL ice-cold 1x PBS (Lonza) twice. 9 mL of PBS was added to the flask and cells were detached by adding 1 ml of 10x Trypsin-EDTA (Sigma-Aldrich) and incubated at 37 °C for 5 minutes. Trypsin-EDTA was then inactivated with 15 mL of DMEM 10% FBS and cells were incubated at 37 °C for 5 minutes. The cells were then pelleted at 300 g for 3 minutes and the supernatant was aspirated. For the experiment, 1 μL of Calcein-AM and 1 μL of ethidium homodimer-1 were added to one millilitre of washed HEK293T cells and incubated on ice for 25 minutes. The cells were then pelleted at 500 g for 5 minutes at 4 °C and resuspended in 1x PBS, and brought to a concentration of 250 cells per μL for 1x loading and 1250 cells per μL for 5x loading. The cells were then mixed 1:1 with a solution of 1x PBS + 30% (v/v) OptiPrep (Sigma-Aldrich) for encapsulation. To assess the sorter's performance when processing low viability samples (1:1 dead/alive), half of the HEK293T cells were treated with 0.25% (w/v) IGEPAL CA-630 (Merck) for 15 minutes on ice. The dead and alive cells were then pooled, stained and prepared for encapsulation as previously explained.

Wild-type mouse embryonic stem cells (E14Tg2a wild-type, a generous gift from Prof. Austin Smith, Living Systems Institute, University of Exeter) were cultured in 2i+LIF medium (DMEM/F-12 without L-glutamine (Gibco) and Neurobasal medium without L-glutamine (Gibco) in a 1:1 ratio, supplemented with 0.1% sodium bicarbonate (Gibco), 0.11% Bovine Albumin Fraction V Solution (Gibco), 0.5x B27 supplement (Gibco), 1x N2.BV (Cambridge Stem Cell Institute), 50 μM 2-mercaptoethanol (Gibco), 2 mM L-glutamine (Gibco), 1x Penicillin-Streptomycin (Gibco), 12.5 μg/mL human insulin

recombinant zinc (Gibco), 20 ng/mL leukaemia inhibitor factor (Cambridge Stem Cell Institute), 3 μM CHIR99021 (Cambridge Stem Cell Institute), 1 μM PD0325901 (Cambridge Stem Cell Institute)) at a cell density of ~8000 cells/cm². To passage the cells or generate a single cell suspension, the cells were treated with Accutase (Merck) for 3 minutes at 37 °C and subsequently washed with 10x the volume wash medium (DMEM-F12 + 1% BSA), centrifuged at 300 g for 3 min and then resuspended in either culture medium or 1x PBS (Lonza). For downstream experimental procedures, the cells were processed similarly to the HEK293T cells.

### Preparation of mESC and HEK293T nuclei

The cells were cultured and harvested as described in the previous paragraph. The nuclei suspension was obtained following the Nuclei EZ (Sigma-Aldrich) preparation guidelines applied to cells resuspended in PBS (Lonza) and stained for 20 minutes on ice by supplementing 1 μL of Vybrant DyeCycle Green DNA stain (Thermo Fisher Scientific) to the nuclei resuspended in 1x PBS (Lonza), 7.5% OptiPrep (Sigma-Aldrich) and 0.04% BSA (Thermo Fisher Scientific).

### mESC and HEK293T nuclei or whole cell species mixing loading

The cell solution was loaded at 125 HEK293T cells/μL and 125 mESCs cells/μL or 125 HEK293T nuclei/μL and 125 mESCs nuclei/μL.

### Paraformaldehyde fixation of HEK293T cells

Samples were harvested as described above and washed twice in PBS (Lonza) and re-suspended in 4% paraformaldehyde (Sigma-Aldrich) for 20 minutes at room temperature. The samples were then washed twice in PBS (Lonza) and filtered using a 40 um cell strainer (Greiner). The cells were counted on a haemocytometer and loaded at a concentration of 250 cells/μL.

### Preparation of cell suspensions from the developing mouse brain at E10.5

For animal maintenance, we inspected animals daily and those with any sign of health concerns were culled immediately by cervical dislocation. All experimental mice were free of pathogens and were housed on a 12–12 h light-dark cycle and they had unlimited access to water and food. Temperature in the facility was controlled and maintained at 21 °C, with a humidity of 40%. Mice for post-implantation embryo recovery (CD-1 females and males from Charles River were acclimated for 1 week prior to use) were utilised from 6 weeks of age. Females and males were mated for up to five days or until a plug was found; inspection for vaginal plugs was performed daily. Females were culled by cervical dislocation at the appropriate time point to obtain embryos of the correct age. The day on which a vaginal plug was found was scored as E0.5. Neural embryonic tissue was retrieved by embryo dissection carried out in M2 medium (Sigma). Mouse embryos were carefully collected by cutting open the decidua and the yolk sac. To collect the neural tissue, the pial meninges were removed and the head was cut above the eyes to collect the forebrain, midbrain and part of the hindbrain.

The collected tissue was quickly washed in PBS twice and centrifuged for 5 min at 0.2 x g prior to dissociation with 500 μL of TrypLE Express (Gibco) at 37 °C. The tissue was allowed to dissociate for 15 min, with pipetting for 10–15 times (avoiding bubble formation) every 5 min to help tissue breakdown. If clumps of cells persisted, dissociation was continued for an additional 5 min. Dissociation was halted by adding 2 mL of basal DMEM medium supplemented with 15% of foetal bovine serum. The cell suspension was filtered through a 40 um cell strainer (Greiner), centrifuged, washed once with 1x PBS, centrifuged again and resuspended again in a small volume of PBST (PBS with 0.02% Tween-20) and stored on ice until encapsulation. After the PBS wash, a small aliquot of the cell suspension was used to assess viability by mixing it in a 1:1 ratio with Trypan Blue (Sigma) and

quantified using a haemocytometer. The cells were then incubated for 3 hours at room temperature, after which they were stained with Calcein-AM (Thermo Fisher Scientific) following the manufacturer's instructions, and viability before encapsulation was assessed on a haemocytometer. The cells were then re-suspended in 1xPBS (Lonza) + 15%(v/v) OptiPrep (Sigma-Aldrich) before encapsulation.

### General description of the microfluidic encapsulation, sorting and picoinjection

A detailed protocol[28] for co-encapsulation of cells and barcoded beads was used as a reference for droplet generation. We used high surfactant concentrations for droplet generation and picoinjection (5% RAN fluorosurfactant), as described previously in protocols requiring high-temperature incubation[37,72]. Here we present step-by-step guidelines for performing cell encapsulation and droplet sorting:

1. First, three 2.5-mL or 5-mL glass syringes (SGE) were filled with 5% (w/w) 008-FluoroSurfactant (RAN Biotechnologies) in HFE-7500 (3 M) and connected to oil inlets (Supplementary Fig. 1D, photograph e).
2. Next, three pieces of polyethylene tubing (I.D. 0.38 mm, O.D. 1.09 mm, Portex, Smiths Medical) were connected to two 1-mL gas-tight syringes (SGE or Hamilton) and filled with PBS (Lonza). The tubing was manually filled with PBS, and a small, 1 cm-long air bubble was left at the end tip of each tubing. The aqueous solution for buffer droplets was 1x First Strand buffer (Invitrogen), 4.2 mM DTT (Invitrogen). The aqueous solution for the lysis mix was: 120 mM Ultrapure Tris-HCl (pH 8, Thermo Fisher Scientific), 3.15 mM dNTPs (each, Invitrogen), 0.6% (v/v) IGEPAL-CA630 (Sigma-Aldrich), 7.2 U/ml, Thermolabile proteinase K (NEB).
3. The lysis mix and buffer droplets mix were manually aspirated into the tubing, and the small air bubble provided a separation between the reagents and the PBS (Lonza) buffer.
4. Then, 150 μL of cell suspension (in 1x PBS (Lonza) + 15% OptiPrep (Sigma-Aldrich)) was manually aspirated into the cell loading tip pre-filled with mineral oil (Sigma-Aldrich). Detailed fabrication protocol of cell delivery and droplet collection tips is provided in a recent article[37].
5. Next, all three tubings and the cell chamber with cell suspension were inserted to the corresponding inlets of the droplet generation chip (Supplementary Fig. 1D, photograph f).
6. The droplet collection tip[37] (Supplementary Fig. 1C) and the long outlet tubing were connected to the positive and negative outlets, respectively (Supplementary Fig. 1D, photograph g). The position at the end of the negative outlet tubing was adjusted and placed around 5 cm below the stage of the microscope in order to obtain the desired flow resistance ratio between the positive and negative channels.
7. Finally, in dark conditions, the barcoded beads were aspirated into the tubing which was connected next to the microfluidic device (Supplementary Fig. 2A, photograph h). The inDrop barcoded beads were prepared prior to the experiment according to the inDrop protocol[28], with the inDrop v3 oligonucleotide barcoding scheme[29]. Just before aspiration, the beads were washed three times in 10 mM Ultrapure Tris-HCl (pH 8, Thermo Fisher Scientific), 0.1 mM EDTA (Invitrogen) and 0.1% Tween-20 (Fisher Scientific) and resuspended in 55 mM Ultrapure Tris-HCl (pH 8, Thermo Fisher Scientific), 0.1% (v/v) IGEPAL CA-630 (Merck), 75 mM KCl (Invitrogen), 0.05 mM EDTA (Invitrogen), 0.05% Tween-20 (Fisher Scientific). The lysis mix was as follows: 120 mM Ultrapure Tris-HCl (pH 8, Thermo Fisher Scientific), 3.15 mM dNTPs (each, Invitrogen), 0.6% (v/v) IGEPAL CA-630 (Merck), 7.2 U/mL Thermolabile proteinase K (NEB).
8. The solutions were injected in the in-line microfluidic device using neMesys pumps (Cetoni) and sorted with the following flow rates:

150 μL/hr for the lysis mix, 150 μL/hr for the cell mix, 60 μL/hr for the beads, 700 μL/hr for the main oil, 2500 μL/hr for the spacing oil to generate droplets with average volume 1.3 ± 0.08 nL with a throughput of around 75 droplets per second that falls in the range of 10-100 Hz for passive droplet generation outlined in the original inDrop protocol[14]. Additional flows of 15 μL/hr for aqueous phase and 30 μl/hr for oil were applied to generate buffer droplets.

9. Next, the fluorescence was recorded in high throughput and droplets were sorted according to the set threshold. After stabilisation of a droplet flow, the syringe with mineral oil was removed from the collection tip outlet tubing and the droplet sorting began. A short 1-ms-long 1 V pulse was delivered with a delay of 3.5 ms to the microfluidic device by 'salt electrodes' filled with 5 M NaCl solution and, as a result, highly fluorescent droplets with single cells were derailed to the collection channel for positive 'hits' (b). The LabVIEW algorithm for fluorescence detection was developed previously for FADS sorting[71]. The duration and delay of pulse can be modified according to the flow rates and the desired throughput of the sorting (usually around 70 Hz), depending on the droplet generation rate. The sorting device was used only once for each experiment since the droplet generation was performed on the same chip.

10. The positive and buffer droplets were collected in a collection chamber pre-filled with mineral oil and incubated at room temperature (23 °C) for 25 minutes. The barcodes were then solubilized from the bead via UV exposure using a high-intensity UV inspection lamp (UVP, Analytik Jena) for 7 minutes.

11. The collection chamber was then immersed in a water bath placed at 70 °C for 10 minutes and was subsequently immersed in an ice-cold recipient (half part water and half part ice) for 5 minutes after the incubation was finished.

## Picoinjection of RT mix

Before starting the picoinjection of droplets containing single-cell lysates, the electrode sections (Supplementary Fig. 1B) of the devices, were pre-filled with filtered 5 M NaCl as previously described. The picoinjection chip was filled with 5% (w/w) 008-FluoroSurfactant (RAN Biotechnologies) in HFE-7500 (3 M) using a pre-filled 2.5-ml glass syringe (SGE) connected to a piece of tubing (I.D. 0.38 mm, O.D. 1.09 mm, Portex, Smiths Medical). The reaction mix was primed, and the tip containing the emulsions (with fluorinated oil evacuated by pushing the glass syringe until the emulsions reached the exit of the tip) was primed and connected to the device. The droplets were then re-injected in a pico-injector device and coalesced with a RT solution at 1:1 ratio of flow rates. The variation in droplet sizes caused uneven spacing during droplet reinjection which occasionally resulted in the formation of smaller 'orphan' droplets containing RT buffer (as observed in Supplementary Movie 2). Formation of these buffer droplets was rare and did not impact the reaction inside the droplets with single-cell lysates. Overall, 98% of droplets were picoinjected correctly using this picoinjection geometry. The RT solution was 1.8x First Strand Buffer (5x FS, Invitrogen), 2.52 mM MgCl2 (Ambion), 9 mM DTT (Invitrogen), 3.6 U/μL RNaseOUT (Invitrogen), 24 U/μL SuperScript III (Invitrogen) and the droplets were incubated for 2 h at 50 °C followed by 70 °C for 15 min. For testing the effect of molecular crowding on RNA capture, 13.5% PEG8000(NEB) was added to the RT solution. For RT conditions using the Maxima H-, the following mixture was used: 1.8x RT buffer (Invitrogen), 2.5 mM MgCl2, 2 U/μL RNaseOUT (Invitrogen) and 18 U/μL Maxima H- RT (Invitrogen) and the RT incubation temperature was switched to 42 °C for 2 h. The flow rates were the following: 200 μL/hr for the droplet mix, 200 μL/hr for the RT solution, 40 μL/hr for the spacing oil and 400 μL/hr for the main oil. The libraries were then prepared as per the inDrop

library preparation protocol and sequenced on a Nextseq 75 bp High Output Illumina kit (Read 1: 61 cycles, Index 1&2: 8 cycles each and Read 2: 14 cycles).

## Processing of mouse C57BL/6 PBMCs for cell-type enrichment

Frozen Splenocyte vials from normal adult C57BL/6 mouse (#SC-M5540-57, Caltag medsystems, sex unknown) were thawed in a water bath at 37 °C and immediately pipetted in 13 ml of pre-warmed (37 °C) 1xPBS (Lonza), 10% FBS. The mix was spun down at 300 g for 5 min at 4 °C and washed twice with 1x PBS 2% FBS. The cells were then strained through a 50 μm cell strainer and 1 million cells in 100 μL were incubated with 10 μL of Fc receptor block (Miltenyi Biotec) for 10 min on ice. 2 μL of PE-anti IgM (1 test unit in 116 μL, #130-116-312 Miltenyi Biotec), CD19 (0.3 μg in 116 μL; 2.6 ng/μL final concentration, #130-112-035, Miltenyi Biotec) and CD45R antibodies (0.3 μg in 116 μL; 2.6 ng/μL final concentration, #130-110-846, Miltenyi Biotec) were added to the mixture and the cells were further incubated on ice for ten minutes. The cells were then washed two additional times with ice-cold PBS (Lonza) and counted and encapsulated using the native inDrop conditions. The light source used for FADS-sorting of B-cells was a diode-pumped solid-state (DPSS) OBIS 561 nm 50 mW laser (Coherent), and the emission was captured through one notch filter 561 nm and two bandpass 593/40 nm filters. The droplets from each sorting channel were imaged under an EVOS FL fluorescence microscope. The libraries were then processed as described in the previous paragraph.

## Processing of mouse embryos at E8.5 for nascent RNA sequencing using 5EU-seq

The embryos were recovered as described in the previous sections and placed in IVC1[73] media for 3 hours in presence of 500 μM 5EU (5-ethynyl-uridine, Sigma-Aldrich). The embryos were cut into smaller pieces using sharp blades, and were dissociated with TrypLE (Thermo Fisher Scientific) for 5 minutes at 37 °C. The reaction was quenched using DMEM/F-12 (Thermo Fisher Scientific) supplemented with 10% FBS and the cells were fixed and pre-processed as per the scEU-seq protocol[50]. The cells were then resuspended in PBS (Lonza), counted and processed through the spinDrop workflow without fluorescence-activated droplet sorting. The downstream library preparation protocol after de-emulsification follows the methodology described in the scEU-seq methodology, however, the final PCR amplification was performed as per the inDrop protocol, to account for differences in sequencing adapters.

## Computational analysis

The bcl files were converted to fastq using bcl2fastq (Illumina) and quality controlled using FastQC[74] and demultiplexed using Pheniqs[75]. For benchmarking of HEK293T cells, the biological read from the 10x Chromium v2 dataset was trimmed to 61 base pairs to match the length of the reads for the libraries prepared with spinDrop and inDrop. Both mouse (GRCm38, ensembl 99 annotations) and human (GRCh38, ensembl 99 annotations) genomes were indexed using STAR[76] (the number of mapped reads for the embryonic mouse brain at E10.5 (72.2%) and 5EU-seq (60.9% for the non-nascent part and 63.4% for the nascent RNA)) and gene expression matrices were generated using zUMIs[44]. Gene names and biotypes were queried from bioMart[77] and downstream integration of datasets and tertiary analysis were performed using Seurat v3[46] package. For downsampling measurements and intronic and exonic repartition calculations, the matrices for each coverage were obtained from the dgecounts rds object generated from zUMIs. Quality control on cell barcodes was achieved using DropletQC v1.0[19] on unfiltered matrices generated using zUMIs. For integration of the 10x Chromium v1[45], spinDrop and sci-RNA-seq3[11] mouse brain datasets, the *FindTransferAnchors* function

from Seurat v3 was used to create a shared embedding. To transfer the annotations from the 10x Chromium v1 to the other datasets, the Louvain algorithm was used (*FindClusters* function) to define clusters in the shared embedding. The label was then transferred to the shared embedding using a maximum Pearson correlation coefficient calculation between the average expression of the 10x Chromium v1 dataset labels and the corresponding cluster ID from the shared embeddings. Differential gene expression analysis was then conducted using a Wilcoxon rank sum test (*FindMarkers* function). RNA velocity analysis was performed using the scVelo package[39] and trajectory inference was performed using CellRank[52]. The equivalent embryo inDrop dataset[51] at E8.5 was randomly downsampled to 936 cells to match the size of the datasets.

### Statistics and reproducibility
For each sample, the sample size was determined to cover each cell-type with at least ten cells per cluster to enable sufficient downstream statistical power for differential expression analysis, for example. Emulsions were collected in batches of ~1000 cells to keep barcode collision rates low. For proof-of-concept experiments with cultured cells with low estimated variability (HEK293T cells and mESC cells), the sample size was ~500 cells. For more complex datasets with multiple cell-types (PBMCs, mouse brain sequencing and 5EU-seq), the sample size was between 1000 and 2000 cells.

Low quality cells were filtered out of the datasets where appropriate in datasets generated using alternative technologies (10x Chromium mainly) according to protocol-specific instructions, based on low gene and UMI counts and abnormal fraction of reads mapping to mitochondrial RNAs. For the 5EU-seq dataset, low-quality empty droplet cells were excluded from the dataset.

Datasets were benchmarked against existing datasets generated using 10x Chromium, inDrop and sci-RNA-seq3 in order to estimate cross-method reproducibility. For spinDrop replicate analysis, two independent replicates were generated from HEK293T cells, yielding reproducible metrics across replicates.

There was no allocation of test subjects for any experiments, thus randomisation was not applicable to our study. For experimental setup and downstream analysis, the researchers needed to know samples, cell types and protocols. No blinding was performed. Data analyses were performed by unbiased software programs/algorithms whenever possible. Unless stated otherwise, differential expression analysis was performed using a two-tailed Wilcoxon rank sum test.

### Reporting summary
Further information on research design is available in the Nature Portfolio Reporting Summary linked to this article.

## Data availability
The sequencing data are available at the following accession number GSE208156. The 1:1 3T3 and HEK293T mixture 10x Chromium v2 dataset used for benchmarking HEK293T cells is available on their website in the 'Datasets' category (1k 1:1 Mixture of Fresh Frozen Human (HEK293T) and Mouse (NIH3T3) Cells [https://www.10xgenomics.com/resources/datasets/1-k-1-1-mixture-of-fresh-frozen-human-hek-293-t-and-mouse-nih-3-t-3-cells-2-standard-2-1-0]). The mouse PBMC dataset generated using the 10x Single Cell Immune Profiling (v2) kit is available on their website in the 'Datasets' category (Integrated GEX and VDJ analysis of Connect generated library from mouse PBMCs [https://www.10xgenomics.com/resources/datasets/integrated-gex-and-vdj-analysis-of-connect-generated-library-from-mouse-pbm-cs-2-standard-6-0-1]). The sci-RNA-seq3 E8.5 mouse dataset was obtained from the TOME dataset [https://shendure-web.gs.washington.edu/content/members/cxqiu/public/nobackup/tome_

summary_data/mm/seurat_object_E8.5b.rds]; similarly to the E10.5 mouse dataset [https://shendure-web.gs.washington.edu/content/members/cxqiu/public/nobackup/tome_summary_data/mm/seurat_object_E10.5.rds]. The 10x v1 mouse brain dataset was downloaded from SRA with accession number PRJNA637987. The inDrop mouse organogenesis dataset at E8.5 is available on GEO with accession number GSE189425. The designs in Supplementary Fig. 1A and B can be found in our repository DropBase [https://openwetware.org/wiki/DropBase:Devices]. Source data are provided with this paper.

## Code availability
Code for sorting and bioinformatic computation is available at https://github.com/droplet-lab/spinDrop[78].

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

## Acknowledgements

J.D.J. received scholarship support from the BBSRC, T.S.K. was supported by EU H2020 Marie Skłodowska-Curie Individual Fellowship (MSCA-IF 750772), A.L.E. was supported by the Cambridge Trusts and the EU H2020 Marie Curie ITN MMBio and T.N.K. by an AstraZeneca studentship. M.T. was supported by the International Centre for Translational Eye Research (MAB/2019/12) project, which was carried out within the International Research Agendas programme of the Foundation for Polish Science, co-financed by the European Union under the European Regional Development Fund. This work was supported by the EU Horizon 2020 programme (ERC Advanced Investigator Awards to F.H., 69566 and M.Z.G., 669198), the Wellcome Trust (WT108438/C/15/Z to F.H. and 207415/Z/17/Z to M.Z.G.) and the NIH (Pioneer Award to M.Z.G., DP1 HD104575-01). The authors would like to thank the members of the Hollfelder laboratory for their feedback. We thank Dr. Anna Alemany for help and suggestions for the data analysis.

## Author contributions

J.D.J., T.S.K. and F.H. conceptualised the study. T.S.K. and J.D.J. developed and optimised the droplet microfluidic workflow. J.D.J. developed and optimised the molecular workflow. J.D.J., A.L.E. and T.N.K. retrieved the cultured cells. G.A., C.H. and J.D.J. retrieved and processed the mouse embryos. J.D.J., T.S.K. and D.B.M. performed the encapsulations. J.D.J. performed library preparation and sequencing. J.D.J. and M.T. performed downstream analysis of sequencing results. J.D.J., T.S.K. and F.H. wrote the manuscript, with input from all authors. F.H., G.M.F., S.T. and M.Z.G. supervised the work.

## Competing interests

J.D.J., T.S.K. and F.H. are inventors on a patent application (PCT/GB2021/052111) related to the methods presented in this publication and submitted on behalf of the University of Cambridge via its technology transfer office, Cambridge Enterprise. S.A.T. is a Scientific Advisory Board member of Foresite Labs, Qiagen and Element Biosciensces, and a co-founder and equity holder of TransitionBio and EnsoCell. The remaining authors declare no competing interests.
