## [Peer Review File · Nature Communications]

REVIEWER COMMENTS

Reviewer #1 (Remarks to the Author):

This manuscript very nicely describes a technique to enrich for high quality RNA, while still maintaining high-throughput single cell analysis, no easy task. The characterization of their system is thorough and expansive, testing live cells and the very difficult PFA-fixed cells. The sorting for viability after co-encapsulation is very clever and I have no doubt it will contribute greatly to single cell analysis field and attract attention from companies such as 10X genomics. The reagent addition via dielectrophoretic picoinjection is also very interesting with an even broader range of applicability of interest from others in the microfluidics field. All around, I think the manuscript is wonderful, I just have several comments below mainly out of interest generated as I read the manuscript (which has data that is well analyzed and is also well written).

Can you give an example image of the threshold for fluorescence detection over negative background (line 161) and what an arbitrary threshold may look like (line 764). Also, In Figure 2B, there seems to be variability in fluorescence per cell, how does your software/thresholding handle this?

Is software for detection written in house? Could you make scripts open source?

For line 164, I admire the use of dielectrophoresis for "pulling out" droplets and I have seen the supplementary video demonstrating the process (Supplementary Video 1) and want to know the refractory time and reliability of pulling out droplets. For instance, how many times does the system miss pulling out a "good" droplet due to failure of force generated by dielectrophoresis? And does this vary/ is it sensitive to size differences in droplets it is trying to pull? Also on that note, I see the droplets are on average 1.3 nL (line 977), however can I see a size distribution of this? Also, after looking at Supplementary Video 1, it seems like the applied dielectrophoretic force cause droplet deformation, does this ever result in droplet breaking? What is the optimization done here to provide a strong enough force for sorting, but not strong enough to break the droplet?

In extended data Figure B, you are missing the label "4"

Labeling directly onto Figure 1 would be appreciated.

In your supplementary Video 2, there is a "miss" where the picoinjection droplet misses the droplet and forms a smaller droplet, not containing a cell. How often does this event occur? What do you do with these miss-injected droplets? What and how is the timing if picoinjection controlled? There also seems to be a variety of droplet sizes in this video, can you remark on this?

5% 008-FluoroSurfactant seems like a lot (line 1003), was this optimized? How did you arrive at this concentration and surfactant?

Line 1015, how long at 42OC?

Although I appreciate the 1:1 live:dead cell experiment, on line 930, you mention viability analysis on aliquots of samples and I am curious how the trypan blue viability numbers align with the sorting numbers you see via FADS?

Step 11, line 911, could you remark on integrity of RNA after UV exposure?

Line 206, and Figure 2C, how is predicted value calculated?

Line 608, Background calculated via zUMIs pipeline: proportional to reads matching to a cell barcode, how does it relate to ambient RNA (Caglayan et al. 2022, Neuron)

Line 260-264, I very much appreciate the optimization across 3 methods (Figure 3A), can you elaborate on the heat denaturation (duration and temperature) please.

PFA looks like it captures both nuclei and cytoplasmic RNA well! Could you include this in a small sentence somewhere?

Line 377, there is mention of cortical maturation markers, however there are also some mitochondrial genes highly enriched (Figure F), could this be an indicator of cell integrity?

What is your reverse cross-linking protocol?

For Figure 5, is there a way to present a UMAP/reduced dimension with just nascent versus non-nascent then the combination of the two? This would be to better understand how the nascent transcripts dictate the broader relative situation of cells relative to one another.

Is there a way for you to show that the RNA velocity trajectories are largely improved in spinDrop due to increase in mapped intronic reads/intersections? Mainly to isolate the exact features/advantages provided by spinDrop here.

The discussion is written well relative to the technical side of spinDrop, however, can there be more mention on the biological relevance of the findings from Figure 4 and 5? Perhaps on importance on non-coding RNAs (with examples like Gm8292, Mir703, CT010467.1) on the process of neural development? Or a little more elaboration on Shh in organogenesis?

Reviewer #2 (Remarks to the Author):

In this manuscript, Jonghe et al. extensively optimized the single-cell profiling platform (inDrop-seq) to improve its efficiency and reduce the background noise significantly. They applied the technique for profiling cultured cells/nuclei and mouse embryonic tissues, and tested its compatibility for profiling nascent RNA transcription during mouse organogenesis. As an expert in the field of genomic technique development, I am generally excited about the significant improvement of the new approach and appreciate the authors' efforts. The following are several comments that should be fixed before the publication of the method:

1. The authors use the FADS approach to remove doublets by applying an upper threshold on fluorescence. One potential concern is that this may introduce bias to the profiled cell population. This should be validated. In addition, it is possible that cells in the S/G2/M phase could be filtered out in this step.

2. For figure 2E, it would be good for the authors to compare the transcripts detected per cell for the immune cells profiled by SpinDrop-seq and 10x.

3. What is the throughput limit of this technique? One potential concern is that the sorting process could be the speed limiting factor.

4. While removing dead cells during encapsulation could be helpful, how does this compare with the typical strategy (sorting live cells before the 10x experiment)? Also, is it possible to FACS sort the droplet with encapsulated cells?
5. In figure 3A, what is the heat denaturation step? More details are needed in the legend.
6. How does the technique's efficiency compare with the latest 10x technique (v3)?
7. In Figure 3C, why do the mouse cells show relatively low purity? How does the purity compare with the 10x technique?
8. In Figure 4E, why the neural crest cells are missing in spinDrop? And the authors should compare the efficiency of the technique with 10x and sci-RNA-seq3 in this experiment.
9. How does the proportion of EU reads change across different cell types? This should be quantified.
10. It would be important that the authors comment on any potential limitations or concerns of the technique. Also, the author should provide detailed step-by-step protocol and computation script for the broad application of this technique.

Reviewer #3 (Remarks to the Author):

spinDrop is proposed as an advancement for low cost, open source scRNA-seq with results comparable to that of 10X Genomics. In this technique, droplet sorting allows selection of those droplets that contain cells of interest, removing a significant amount of background noise from empty cells, dead cells, and other issues. This is followed by picoinjection to introduce reverse transcription (RT) reagents to select cells, allowing harsher lysis methods while still achieving RT in-droplet, similar to inDrop and 10x genomics protocols. Overall, the manuscript is straightforward. The authors present their technological additions, test them separately to confirm that they are achieving their goals (cell sorting/RT picoinjection), optimize parameters, and compare to other platforms.

I do not recommend this manuscript for publication as the work lacks novelty and does not seem to be of broad applicability.

Major concerns:

1. The microfluidic schema proposed here is not conceptually new; see refs. 37, 58, <https://doi.org/10.3390/bioengineering9110674>
2. The use of 2 different microfluidic devices make the workflow less user-friendly. The microfluidic driver setup is also complicated and unavailable in most laboratories. In particular, microfluidic picoinjectors and sorters are more difficult to fabricate than the passive droplet generators that are used for most droplet-based scRNA-seq (InDrop, Drop-seq, 10x Genomics). On-chip electrodes also have lower efficiency in fabrication electrical contact is easy to break during handling. Pairing two such devices make the workflow twice as likely to fail.
3. The authors fail to discuss the throughput for the spinDrop setup: how many cells are processed in a typical experiment, using spinDrop? From fig. 4D, the cell numbers look to be too low to be of practical use (2,472 cells for spinDrop with FADS sorting and 802 cells for spinDrop without FADS sorting). I was unable to tell what the cell numbers were in Fig. 2E or Extended Data Figure 4A,B. Is this method scalable?

4. Both droplet sorters and picoinjectors take longer to run (sort, picoinject) than passive droplet generators. What are the batch effects introduced from running ~10k cells in an experiment? Note that high cell numbers are typical in most single cell RNA-seq experiments (cell atlas'ing experiments). In any case, some experiments should be performed in replicates to demonstrate reproducibility.

Minor concerns:

1. When testing FADS, calcein-AM is used to get droplets with single, viable cells at a tested rate of (96.1%, n= 51). N=51 appears to be the single image presented in Figure 2B.

Multiple images should have been processed for this test, including from different collections through FADs. A single image could be the result of an above average sorting run, is this 96.1% value consistent across multiple FADS run at different times?

2. It is unclear why a species-mixing experiment was used to calculate the proportion of reads mapped to a cell barcode. What is the proportion of mapped reads in other spinDrop experiments, e.g., on embryonic mouse brain at E10.5 (Fig 4) and 5EU-seq (fig. 5)?

3. It is unclear why single cells vs. single nuclei species mixing experiments yielded different doublet rates (2.9% for cells vs. 6.1% for nuclei, Extended Data Figure 4A, B). Was the target Poisson loading concentration different in the two experiments?

4. The Wilcoxon rank sum test (lines 284-288) result could be made clearer. Out of the 690 differentially expressed genes, how many were upregulated in spinDrop vs 10x? What does this mean?

Responses to the reviewers' comments for manuscript NCOMMS-23-00685-T

Reviewer #1

This manuscript very nicely describes a technique to enrich for high quality RNA, while still maintaining high-throughput single cell analysis, no easy task. The characterization of their system is thorough and expansive, testing live cells and the very difficult PFA-fixed cells. The sorting for viability after co-encapsulation is very clever and I have no doubt it will contribute greatly to single cell analysis field and attract attention from companies such as 10X genomics. The reagent addition via dielectrophoretic picoinjection is also very interesting with an even broader range of applicability of interest from others in the microfluidics field. All around, I think the manuscript is wonderful, I just have several comments below mainly out of interest generated as I read the manuscript (which has data that is well analyzed and is also well written).

We thank the reviewer for their input and share their enthusiasm about the spinDrop methodology. We especially thank the reviewer for the useful instructive comments that we have addressed below and in the manuscript (shown in red).

Can you give an example image of the threshold for fluorescence detection over negative background (line 161) and what an arbitrary threshold may look like (line 764). Also, In Figure 2B, there seems to be variability in fluorescence per cell, how does your software/thresholding handle this?

Unfortunately, the version of the algorithm we used does not save the samples of raw recordings, but the example of arbitrary threshold using both amplitude and the droplet residence time (width of the peak) is presented in the extended data fig. 2 and we show a rendering of a typical recording, digitised from a snapshot of the sorting interface in Figure R1.1a. The baseline frequency of the signal represents each travelling droplet. The droplet containing a green fluorescent cell will pass the sorting

threshold based on amplitude (dotted line), and, if the area of the signal aligns with other singlets, the droplet is sorted (similarly to FACS). The variability of fluorescence signal results from the heterogeneity of the cell size because of differences in cell-cycle stage, mainly; or more broadly intracellular esterase content, although this is arguably hard to quantify. However, the thresholding on fluorescence is kept broad and is solely applied to remove the background empty droplets or droplets with damaged cells, or very bright cellular aggregates, which can clearly be distinguished and are presented in Figure 2A. Although our analysis reveals the proportion of cells per cell-type is conserved in the mouse brain dataset for the sample with and without sorting (Figure R1.1b,c); and that cell-cycle phases throughout the mouse brain and B-cell sorting datasets are broadly similar with and without sorting (Figure R1.d,e), which indicates that our sorter performs indiscriminately from cell-size and fluorescence levels, a possible iteration on the current technology would be to implement multi-modal sorting e.g. combining the fluorescence readout with image-based analysis to integrate this information in the subsequent single-cell analysis and increase confidence in the populations sorted. We have added these points of discussion in the main text and discussion section:

- 1) “To verify that our sorting parameters did not significantly affect the population of B-cells profiled (mainly due to their size), cell-cycle phase was profiled for all datasets, revealing that the proportion of cells in each phase was broadly similar between all datasets. This observation refutes that selections were based on cell sizes as the latter varies significantly throughout cell-cycle stages (Extended Data Figure 2F).”
- 2) “Sorting using FADS did not affect cell-type representation (apart from a slight overrepresentation of fibroblasts in the FADS dataset) or the proportion of cell-cycle phase, showing that the method is broadly applicable to cell atlasing and does not affect cell-type representation (Extended Data Figure 4E-G)”
- 3) “Although sorting did not significantly skew cell-type proportions in our profiling endeavours, future implementations of the sorting technology could comprise an image analysis component to more faithfully discard cell doublets from large cells, or even add a phenomic component to the sequencing results, e.g. based on cell size or morphology.”

These data can now be found in the new version of the manuscript in Extended Data Figures 2F and 4E-G.

Figure R1.1 Fluorescence-activated droplet sorting enriches for single-cells. a) amplitude of the green fluorescence signal, arbitrary threshold is indicated with a dotted line (0.8 V), and the area marked in green will contain the signal that may be sorted, depending on the area of the curve. b) cluster distribution for the mouse brain dataset at E10.5 with and without FADS, showing essentially similar cell-type proportions apart from fibroblasts (cluster 1). c) UMAP representation of the datasets represented in b). d) Cell-cycle phase inference for B-cells in the PBMC sorting experiment across datasets, showing that fractions of cells in different cell-cycle phases are essentially equivalent, refuting that size distribution (hence also fluorescence) impacts the sorting process. e) Cell-cycle phase inference for the mouse brain at E10.5 sorting across datasets, showing that fractions of cells in different cell-cycle phases are essentially equivalent, refuting that size distribution (hence also fluorescence) impacts the sorting process.

Is software for detection written in house? Could you make scripts open source?

Indeed, the software for detection was written in-house and we have added the LabVIEW script employed to GitHub.

For line 164, I admire the use of dielectrophoresis for “pulling out” droplets and I have seen the supplementary video demonstrating the process (Supplementary Video 1) and want to know the refractory time and reliability of pulling out droplets. For instance, how many times does the system miss pulling out a “good” droplet due to failure of force generated by dielectrophoresis? And does this vary/ is it sensitive to size differences in droplets it is trying to pull? Also on that note, I see the droplets are on average 1.3 nL (line 977), however can I see a size distribution of this?

In our hands, the sorting process performs robustly, as shown in the results presented in Figure 2C, demonstrating that only a few live cells remain unsorted (Figure 2C). During the experiments, we have not noticed droplets that are not pulled despite being detected. The refractory time of the electronics is very short and does not influence the sorting process - a field-programmable gate array (FPGA) used in our system has a sampling rate of 200 kHz (which means that sorting operations can be in theory executed every 5 microseconds). Practically the throughput is limited by the duration of the high-voltage pulse that usually lasted 1 ms.

We have run an image analysis of the videos with droplets and calculated the relative standard deviation (RSD) of the droplet area (in the 2D image) to be equal to 4.2%, which can be translated to the RSD of droplet volume to be around 6.3% (assuming a spherical droplet shape). We have amended the droplet volume in the text to 1.3 ± 0.1 nL. This size variation of droplets generated in splndrop is slightly larger than for droplets typically formed by microfluidic devices (standard microfluidic flow-focusing junction generates emulsions with polydispersity index typically $< 1\%$ ¹). This is due to the imbalance in droplet generation because of the barcoded bead packing process, which negatively affected droplet monodispersity compared to purely aqueous droplets (a general limitation of single-cell technologies). In our experience, it is not the variation in droplet sizes that affects the throughput of the

sorting, but rather the uneven spacing between the droplets due to uneven bead packing. Reduced solid-support bead sizes would reduce this imbalance and in turn ameliorate throughput. We have added this point of discussion in the manuscript with the following text:

”Sorting throughputs might be increased by using smaller beads that would provide better droplet monodispersity by reducing the droplet volume, or using serial electrodes that can improve sorting speed².”

Also, after looking at Supplementary Video 1, it seems like the applied dielectrophoretic force cause droplet deformation, does this ever result in droplet breaking? What is the optimization done here to provide a strong enough force for sorting, but not strong enough to break the droplet?

We have not observed any droplet breaking in 3 recorded video traces across 60 droplets, and in our hands and according to our sorting output calculations, the sorter performs robustly throughout multiple experiments. However, we agree that sorting of droplets for single-cell genomic assays is more challenging due to the presence of high concentrations of water-soluble surfactants (e.g. Igepal CA-630) in the lysis mix. Under these conditions, the interfacial tension between the droplet aqueous phase and the oil fluorocarbon phase is much lower than normal, which makes the droplet more “deformable” and more difficult to transport to the sorting channel using an electric field. To alleviate this, we made the sorting junction deeper than the droplet generation junction (which required the fabrication of a 3-layer chip). As a result, droplets are spherical at the sorting junction, rather than squeezed, and they are much easier to translocate with an electric pulse. To clarify these optimisations, we have added the following sentence in the methods section:

“Additionally, we implemented a gapped divider at the sorting junction³ that gradually pushes droplets to the outlet channels and minimizes the risk of droplet breaking.”

We have also reviewed our description of the sorting protocol and added a sentence emphasising the importance of adjusting flow resistance in the negative channel (unsorted droplets).

In extended data Figure B, you are missing the label “4”

We thank the reviewer for this comment. We assumed the reviewer asked for Extended Data Figure 1B, which has a label 4, however it may have been harder to spot as it is located at the bottom of the chip as opposed to the other labels that are at the top (Figure R1.2).

Figure R1.2 Indication of label 4 in Extended Data Figure 1B

Labeling directly onto Figure 1 would be appreciated.

We agree that labelling directly onto Figure 1 will assist readers in interpreting the method, and have modified the Figure accordingly, as shown in Figure R1.3.

Figure 1

Figure R1.3 New Figure 1 with inserted labels.

In your supplementary Video 2, there is a “miss” where the picoinjection droplet misses the droplet and forms a smaller droplet, not containing a cell. How often does this event occur? What do you do with these miss-injected droplets? What and how is the timing if picoinjection controlled? There also seems to be a variety of droplet sizes in this video, can you remark on this?

We thank the reviewer for pointing this out. The higher variation in droplet volumes (as discussed in one of the previous questions) resulted in less even spacing of droplets during droplet generation. To better explain this phenomenon, we have added the following text in the methods section of the manuscript: “The variation in droplet sizes caused uneven spacing during droplet reinjection which occasionally resulted in the formation of smaller ‘orphan’ droplets with RT buffer (as visible in

Supplementary Video 2). Formation of these buffer droplets was rare and did not impact the reaction inside the droplets with single-cell lysates. Overall, 98% of droplets were picoinjected correctly using this picoinjection geometry.”

5% 008-FluoroSurfactant seems like a lot (line 1003), was this optimized? How did you arrive at this concentration and surfactant?

During our experiments, we observed that the 2% fluorosurfactant concentration used routinely for droplet assays does not fully prevent droplet merging during the lysis step at 70 °C. We therefore used 5% surfactant in our assay following the instructions found in another protocol - the MaPS-seq assay, a modification of inDrop method for spatial metagenomics of gut microbiome, developed by Sheth *et al.*⁴. This protocol routinely used 5% RAN fluorosurfactant for co-encapsulation of cells and beads and higher concentrations (30%) were used for subsequent emulsion PCR. Because this optimisation worked well in our hands, we decided to use 5% fluorosurfactant throughout our assay. We have added this reference and explanation in the following text added to the methods section: “We used higher surfactant concentrations for droplet generation and picoinjection (5% RAN fluorosurfactant) as described previously in protocols requiring high-temperature incubation^{4,5}”.

Line 1015, how long at 42OC?

We apologise for omitting the incubation time, which was 2 hours, we have modified the manuscript accordingly and also improved the methods section by adding further details.

Although I appreciate the 1:1 live:dead cell experiment, on line 930, you mention viability analysis on aliquots of samples and I am curious how the trypan blue viability numbers align with the sorting numbers you see via FADS?

We thank the reviewer for their comment. We artificially killed the population of HEK293T cells with the non-ionic detergent IGEPAL-CA630 and performed dual staining with Calcein-AM (green fluorescence) and ethidium homodimer-1 (red fluorescence), which confirmed that no cells were alive after treatment (no green

fluorescence from Calcein-AM was detected on the haemocytometer). The “dead” cells were counted using the ethidium homodimer-1 signal and mixed 1:1 with a viable sample (>99% viable, as confirmed by the Calcein-AM stain). The cells were then sorted based on viability and resulting droplets were assessed for encapsulated cell viability on the haemocytometer. Unfortunately, because only green fluorescence is measured during the sort, it is not possible for us to provide metrics for this experiment at the sort level as only viable cells can be counted (and dead cells will be indiscriminate from empty droplets). Therefore we counted the cells contained in the droplets post-sorting using a fluorescence microscope instead of providing sort metrics. We have clarified the text to underline that the results of the sort were assessed using fluorescence microscopy to determine viability of encapsulated cells after the sort:

“To further quantify the potential of our system to extract single viable cells from a challenging sample containing a large proportion of damaged cells, the input population was modified to incorporate a 1:1 ratio of dead and alive HEK293T cells treated with a dual green/red-live/dead stain (Calcein-AM and ethidium homodimer-1, respectively). To induce cell death, a concentration of 0.25% (w/v) IGEPAL CA-630 was added to half of the HEK293T and incubated on ice for 15 minutes. Sorting of a 1:1 mixed population of dead and living cells showed a marked 19-fold enrichment for viable cells from the pool of droplets containing cells, assessed for viability using a fluorescence microscope, with 84.8% of the droplets containing a single viable cell, which surpasses the predicted value of 4.52% without sorting (Figure 2C).“

Step 11, line 911, could you remark on integrity of RNA after UV exposure?

We thank the reviewer for noting this potential bottleneck, which is also found in the inDrop protocol. Although RNA is known to crosslink or degrade (wavelengths <300 nm) with UV exposure, the wavelength used (365 nm) as well as the exposure time and dose (starting the lamp switched off and 7 minutes exposure time) should not significantly affect the quality of the sample⁶, as demonstrated by the high capture rates in this study. However, it is likely that RNA-RNA or RNA-protein cross links may still hinder RNA reverse transcription⁷, however, the reverse crosslinking procedure (proteinase K digestion and heat denaturation at 70°C for 10 minutes) used in

spinDrop should further assist reverse-crosslinking, which may contribute to the gain in gene detection rates observed compared to the native inDrop protocol which also uses UV exposure to solubilize the barcodes in the emulsions. We have modified the manuscript accordingly with the following point of discussion, to describe other methodologies for releasing the barcode using dissolvable beads or the USER enzyme blend:

”Further implementations may also include the use of dissolvable hydrogels⁸ or enzymatically released barcodes⁹ from the solid-support microgels to circumvent potential limitations of the current system, which uses UV-induced barcode release which may reduce RNA capture due to cross-linking.”.

Line 206, and Figure 2C, how is predicted value calculated?

The predicted value of 4.5% for droplets containing a single viable cell was calculated as follows: first, we calculated the fraction of droplets containing a single viable cell with or without dead cells using a λ value = 0.05 for live cells, which amounted to 4.8%. Then we used the same lambda value for dead cells to calculate that 95.1% out of these 48% droplets would not contain dead cells. By multiplying 4.8% by 95.1%, we obtained 4.5% of droplets containing only a single viable cell (Table R1.1).

This calculation is summarised in the table on the following page presenting the most common combination of droplet content with live and dead cells using a 1:1 dead/alive ratio and a total λ value = 0.1 (0.05 for each):

		no cells	1 dead cell	2 or more dead cells
		95.1%	4.8%	0.1%
no cells	95.0%	90.5%	4.5%	0.1%
1 viable cell	4.8%	4.5%	0.2%	0.0%
2 or more viable cells	0.1%	0.1%	0.0%	0.0%

Table R1.1 cell loading statistics computation

We have added these calculations as a new Supplementary Table 1.

Line 608, Background calculated via zUMIs pipeline: proportional to reads matching to a cell barcode, how does it relate to ambient RNA (Caglayan et al. 2022, Neuron)

We agree that the sentence merits clarification. Background in this context relates to barcodes with low coverage that are discarded by the analysis pipeline. Ambient RNA is indiscriminately captured in sorted and unsorted droplets, however, as the capture rates for those are likely to be similar between all droplets (with and without a cell), they would not impact barcode thresholding and selection significantly. The background barcode coverage in the text’s context can result from a combination of multiple factors, such as primer dimers, concatemers, low-quality cells (or more commonly cell debris), and indeed ambient RNA. We have clarified the related figure caption to describe the nature of background as follows:

“Percentage of reads that are mapped to the background, consisting mainly of primer concatemers, ambient RNA and degraded cells or cell debris; and cell barcodes for

inDrop and inDrop with sorting, determined using the filtering statistics from the zUMIs pipeline”.

We have also added a point of discussion in the text regarding ambient RNA captured in the sorted droplets, which is still an artefact present in our current implementation. Methods like SoupX¹⁰ may still be required to remove the uniform ambient RNA signature from the resulting gene expression matrices obtained from sorted droplets. In future implementations, one could possibly reduce the droplet size used for encapsulation significantly to alleviate this problem. Indeed high cell to droplet volume ratios will ensure most of the signal comes from the cell, which leaves less space for ambient RNA to contribute to the single-cell signal in the sorted population:

“However, the approach does not remove ambient RNA from sorted droplets, hence methods like SoupX may still be required to remove ambient RNA co-encapsulated with cells from the dataset. Reduction of droplet size in the future (i.e. high cell volume to droplet volume ratio) may provide a viable route to remove or dampen such artefacts in future implementations.”

Line 260-264, I very much appreciate the optimization across 3 methods (Figure 3A), can you elaborate on the heat denaturation (duration and temperature) please.

We agree that the figure merits further detailing to guide the reader; we have now added the temperature and incubation time (70°C for 10 minutes) in the Figure caption and in the text (Figure R1.4).

Figure 3

A

Figure R1.4 Figure 3A with clarifications regarding incubation temperature and time for the heat denaturation step.

PFA looks like it captures both nuclei and cytoplasmic RNA well! Could you include this in a small sentence somewhere?

We thank the reviewer for noting the efficient RNA capture for PFA-fixed cells, we have further described these gains in a sentence:

“The sample with fixed cells displayed slightly lower gene detection rates, with 1,934 genes with reads mapping to introns and 2,404 genes with reads mapping to exons (Figure 3D). High capture rates for both cytoplasmic and nuclear RNA molecules from fixed samples will broadly expand the number of single-cell methods directly applicable in a high-throughput format, and permit single-cell sequencing after storage, which will be beneficial to clinical samples. In contrast to the newly released 10x Chromium kits ¹¹, spinDrop does not rely on probes for sequencing fixed cells, which should expand the number of species that can be investigated, and yield functional information (e.g. splicing or genotyping) to phenotyping experiments.”

Line 377, there is mention of cortical maturation markers, however there are also some mitochondrial genes highly enriched (Figure F), could this be an indicator of cell integrity?

We have also observed mitochondrial markers in differential expression experiments between spinDrop and 10x datasets. The percentage of mitochondrial read capture is platform and protocol dependent, corroborated for example by recent cross-platform benchmarking efforts where methods that use *in vitro* transcription-based amplification rather than PCR-based, showed higher detection rates of mitochondrial reads.

In addition, it is likely that the proteinase K digestion and heat denaturation may have contributed to enhanced mitochondrial RNA-content release. Because the 10x Chromium method does not employ such harsh lysis conditions, it is likely that our improved lysis protocol may have contributed to higher mitochondrial RNA capture. This is the likeliest explanation, as the nucleus also released a higher proportion of RNA with this lysis method, as corroborated by the enhanced intronic coverage depicted in Extended Data Figure 3A.

Figure R1.5 Percentage of UMIs mapping to mitochondrial genes for HEK293T cells profiled using the 10x Chromium v2, inDrop and spinDrop methods.

However, because the dropletQC metrics show much higher viability in the sorted sample (compared to the unsorted sample), we are confident these observations are linked to protocol-specific capture rather than to lower cell integrity. This nuclear fraction metric is not based on mitochondrial reads as they can be cell-type, species and protocol dependent, but rather on the fraction on unspliced versus spliced reads which help build a “nuclear fraction” ratio that determines alive cells from damaged cells and empty droplets. This result is corroborated by the HEK293T benchmarking effort, where viability was high. In these experiments, the median percentages of UMIs mapping to mitochondrial genes were 10.6% for 10x Chromium v2, 9.3% for inDrop and 14.0% for spinDrop (Figure R1.5). Of note, increased mitochondrial RNA coverage is not necessarily detrimental, as demonstrated by recent advances using mutational signatures detected in mitochondrial RNA to infer cell lineage¹². Increased coverage may thus power these variant calling analysis, which rely on the intrinsic high mutational rates of the mitochondrial genome to infer cell relationships. We have included this new analysis in Extended Data Figure 3B and added a mention in the following text:

“Higher median percentage of UMIs mapping to mitochondrial genes was obtained using spinDrop (10.6% for 10x Chromium v2, 9.3% for inDrop and 14.0% for spinDrop), further underlining protocol-specific capture (Extended Data Figure 3B).”

We have also added lineage tracing as a point of discussion:

“Similarly, increased mitochondrial transcriptome coverage in spinDrop may benefit lineage tracing endeavours¹². “

What is your reverse cross-linking protocol?

For reverse crosslinking, we incubate the droplets at room temperature for 30 minutes to enable proteinase K digestion followed by an incubation at 70°C for 10 minutes, we have added this information to the text and figure captions.

For Figure 5, is there a way to present a UMAP/reduced dimension with just nascent versus non-nascent then the combination of the two? This would be to better understand how the nascent transcripts dictate the broader relative situation of cells relative to one another.

We agree that this would be an interesting analysis. However, integrating these three datasets did not yield significant observable differences (Figure R1.6). We speculate that the subtle differences in capture between new and old transcripts in such a short time frame do not override core differences in cell types guiding dimensional reduction, meaning the old/new/old+new will look relatively similar when projecting on a dimensional reduction space. We therefore turned to MOFA (Multi-Omics Factor Analysis)¹³ to calculate the relative contribution of either normalised matrix (new and old RNA) to the latent factors to further identify unique contributions from the nascent RNA. This analysis revealed that although nascent RNA did indeed contribute less in terms of variance to the integrated latent space (Figure R1.6b), most of the variance for nascent RNA could be found in the second factor (Figure R1.6c). When inspecting the top 10 most variable gene candidates, mitochondrial genes were identified (*Mt1* and *Mt2*), hypothetically underlining the stress undergone by *in vitro* culture of the embryo after dissection. However, morphogenetic transcription factors were also identified, such as *Gata2* and *Tfap2c*, underlining that nascent RNA sequencing could capture unique developmental signatures. We have underlined these new findings in Extended Data Figure 5 G-H and Figure 5I and in the following text:

“To further determine the unique contributions of nascent RNA towards each cell’s transcriptome, multi-omic factorial analysis (MOFA)¹³ was performed on both normalised nascent and non-nascent matrices, which delineated clear contributions of factor 2 to the total variance. Further inspection of the weights contributing to factor 2 highlighted key contributions of transcription factors *Gata2* and *Tfap2c*, underlining core morphogenetic signatures captured via nascent RNA sequencing (Figure 5J and Extended Data Figure 5H-I).”

Figure R1.6 Contribution of nascent RNA to intrinsic variability in the scRNA-seq atlas of mouse organogenesis at E8.5. a) UMAP dimensional reductions of the nascent (new), non-nascent (old) and cumulative non-nascent and nascent matrices (old+new). b) Percent variance in the datasets explained by the non-nascent (old) and nascent (new) normalised matrices identified using MOFA. c) Percent of variance explained for each factor for the non-nascent (old) and nascent (new) normalised matrices. d) top weights for factor 2 showing enrichment for mitochondrial gene and core morphogenetic transcription factors.

Is there a way for you to show that the RNA velocity trajectories are largely improved in spinDrop due to increase in mapped intronic reads/intersections? Mainly to isolate the exact features/advantages provided by spinDrop here.

We agree that additional coverage of intronic regions and overall gain of sensitivity of spinDrop over inDrop could boost RNA velocity predictions. To examine

this further, we explored the datasets in our HEK293T comparative benchmarking experiments (20,000 reads per cell). Because shifts in cell-cycle genes are the likeliest source of variation in a homogeneous sample such as cultured cells [REF], we first projected the cell-cycle scores obtained with the scVelo tool on the three HEK293T objects (inDrop, spinDrop, 10x Chromium v2, Figure R1.7a). This revealed a gradient from S to G2M scores for spinDrop and 10x, across the UMAP1 axis, whereas this was less apparent for the inDrop sample. Based on this analysis, we suspected velocity dynamics could be better captured in spinDrop for cell-cycle genes. Therefore, we ran the scVelo tool in dynamical mode to obtain phase plots for G2M drivers, and obtained clear predictions for induction, steady-state and repression for 10x v2 and spinDrop, but this picture was less clear for inDrop, where the algorithm failed to capture cell-cycle dynamics (Figure R1.7b). This could be explained by most cells having counts only on the spliced x-axis. These findings illustrate how increased sensitivity and intronic coverage using spinDrop can increase accuracy for velocity prediction. We have added these new data in the most recent version of the manuscript, in Extended Data Figure 4A-B and in the following text:

“To determine this, scVelo¹⁴ in dynamic mode was run on the samples. As expected, the core of transcriptional dynamics captured by velocity in the homogeneous HEK293T cultured cells were associated with cell cycle genes. The top 5 most dynamical genes for spinDrop were G2M markers which showed clear induction, steady-state and repression phases, similar to the 10x Chromium data (Extended Data Figure 4A-B). This was less apparent in the inDrop data, which displayed most of the counts along the “spliced” axis, underlining that increased intronic coverage and sensitivity yielded superior dynamical modelling using spinDrop.”

Figure R1.7 Increased sensitivity using spinDrop enables the recovery of dynamical expression of cell-cycle genes in HEK293T cells. a) UMAP dimensional reduction of the HEK293T datasets generated using inDrop, spinDrop and the 10x Chromium v2 protocols; colour gradient indicates cell cycle phase. b) Top G2M cell-cycle candidates displaying dynamical expression in the spinDrop dataset. The analysis shows that increased sensitivity in the spinDrop protocol recovers cell-cycle dynamics which cannot be predicted using inDrop.

The discussion is written well relative to the technical side of spinDrop, however, can there be more mention on the biological relevance of the findings from Figure 4 and 5? Perhaps on importance on non-coding RNAs (with examples like Gm8292, Mir703, CT010467.1) on the process of neural development? Or a little more elaboration on Shh in organogenesis?

We agree that additional validation of the markers could strengthen the manuscript. We have therefore incorporated a novel gene ontology analysis in the

main text to describe the overexpression of core neural markers in the spinDrop in the main text as follows:

”Out of the top 100 markers overexpressed in the neuroblasts profiled using spinDrop, 43 were part of the “Out of the top 100 markers overexpressed in the radial glia profiled using spinDrop, 43 were part of the “Neural system development” gene ontology term (GO:0007399, FDR=3.1e⁻¹⁵), underlining core cell-type specific mechanisms that were absent in the equivalent 10x Chromium dataset.”

Reviewer #2

In this manuscript, Jonghe et al. extensively optimized the single-cell profiling platform (inDrop-seq) to improve its efficiency and reduce the background noise significantly. They applied the technique for profiling cultured cells/nuclei and mouse embryonic tissues, and tested its compatibility for profiling nascent RNA transcription during mouse organogenesis. As an expert in the field of genomic technique development, I am generally excited about the significant improvement of the new approach and appreciate the authors' efforts. The following are several comments that should be fixed before the publication of the method:

We thank the reviewer for their comments and pointing towards the significance of our work, we have aimed to address their concerns by providing more analysis material and description of the work in the manuscript. We have amended the manuscript according to this review using red lettering.

1. The authors use the FADS approach to remove doublets by applying an upper threshold on fluorescence. One potential concern is that this may introduce bias to the profiled cell population. This should be validated. In addition, it is possible that cells in the S/G2/M phase could be filtered out in this way.

We agree with the reviewer that sorting to discard doublets may affect the atlas' content when samples contain different cell types, leading to a large range of fluorescence signal values through staining. In addition, it may be that cells in G2M may be depleted as they are larger in size than cells in G1¹⁵. We have therefore looked at the two datasets in the manuscript that contrast results obtained with and without droplet sorting. First, the mouse brain at E10.5 dataset was inspected. The "Bad cells" low complexity cluster was discarded from both samples (with and without FADS) as it is prominent in the "no FADS" sample and would skew proportional representation. Looking at the global UMAP between both samples and at the cell proportions (Figure R2.1a,b), it appears that cell types are mostly conserved, apart from Cluster 1, which corresponds to fibroblast cells which seemed to be depleted in the sample without sorting. As fibroblasts are relatively average to large in size¹⁶, this finding seems to go against the possibility of excluding larger cells via upper thresholding during the

sorting. We note that, because the “no FADS” sample was processed after the “FADS” sample, it may be that some specific cell-type proportional representations may be skewed due to differing physical properties (i.e. sticking to the loading vessel or different sedimentation rates; or higher/lower cell death rates). In terms of cell-cycle analysis, the FADS-sorted B-cells sample was depleted in the smaller G1 cells compared to the larger G2M and S cells compared to the native inDrop and 10x datasets. This would indicate again that the upper thresholding seems to not be discarding larger cells from the dataset (Figure R2.1c). We however note that these proportions may be highly protocol and sample specific, as indicated by the cell cycle analysis offered on the mouse brain data (Figure R2.1d). Taken together, multiple lines of evidence indicate no depletion of larger cell types despite applying an upper threshold on the fluorescence signal during sorting. We have added these points of discussion in the main text and discussion section:

- 1) “To verify that our sorting parameters did not significantly affect the population of B-cells profiled (mainly due to their size), cell-cycle phase was profiled for all datasets, revealing that the proportion of cells in each phase was broadly similar between all datasets. This observation refutes that selections were based on cell sizes as the latter varies significantly throughout cell-cycle stages (Extended Data Figure 2F).”
- 2) “Sorting using FADS did not affect cell-type representation (apart from a slight overrepresentation of fibroblasts in the FADS dataset) or the proportion of cell-cycle phase, showing that the method is broadly applicable to cell atlasing and does not affect cell-type representation (Extended Data Figure 4E-G).”
- 3) “Although sorting did not significantly skew cell-type proportions in our profiling endeavours, future implementations of the sorting technology could comprise an image analysis component to more faithfully discard cell doublets from large cells, or even add a phenomic component to the sequencing results, e.g. based on cell size or morphology.”

These data can now be found in the new version of the manuscript in Extended Data Figures 2F and 4E-G.

Figure R2.1 Fluorescence-activated droplet sorting does not negatively affect cell-type distributions. a) cluster distribution for the mouse brain dataset at E10.5 with and without FADS, showing essentially similar cell-type proportions apart from fibroblasts (cluster 1). b) UMAP representation of the datasets represented in a). c) Cell-cycle phase inference for B-cells in the PBMC sorting experiment across datasets, showing that fractions of cells in G2M are essentially equivalent, refuting that doublet exclusions through sorting discards larger cells. d) Cell-cycle phase inference for the mouse brain at E10.5 sorting across datasets, showing that fractions of cells in G2M are essentially equivalent, refuting that doublet exclusions through sorting discards larger cells.

2. For figure 2E, it would be good for the authors to compare the transcripts detected per cell for the immune cells profiled by SpinDrop-seq and 10x.

We have computed the differential expression between the sorted B-cells for spinDrop and 10x, which are shown in Figure R2.2. However, this experiment was introduced to demonstrate the sorter (which is encompassed as a theme for Figure 2). Therefore, the improved capture is only introduced starting from Figure 3, and the transcripts captured in the PBMCs would not deviate much from the transcripts obtained with the native inDrop conditions. We initially referred to these experiments as sinDrop (sorting inDrop), but removed this annotation as it may have burdened the reader with too many different technique names.

Figure R2.2 Volcano plot illustrating the differential expression between B-cells from the 10x Chromium and sorting inDrop (sinDrop) dataset. Positive average log₂ fold-change values illustrate higher expression in the 10x dataset, negative values illustrate higher expression in the sinDrop dataset.

3. What is the throughput limit of this technique? One potential concern is that the sorting process could be the speed limiting factor.

We understand this concern, however, we do not see sorting speed as a limiting factor of this technique. The throughput of combined droplet generation and sorting on a single chip (75 droplets per second) in our study falls in the range of throughputs of the passive droplet generation inDrop protocol developed by Klein *et al.*¹⁷, who mentioned in their article that “*The device generates monodisperse droplets that can be varied in the range of 1–5 nl at a rate of ~10–100 drops per second*”. In addition, we showed our system can perform using 5 times the usual cell loading per droplet, due to the capability of discarding doublets at the sorting stage. The demonstration of fast droplet sorting >1 kHz in directed evolution and metagenomic screening^{18,19} campaigns suggests that the throughputs could be increased further by reducing the droplet volume and/or making smaller barcoded beads to improve droplet monodispersity, or implementing serial electrodes that can efficiently sort also larger droplets as demonstrated by Isozaki *et al.*² - we added the description of these possible improvements in the discussion section with the following text:

“Sorting throughputs might be increased by using smaller beads that would provide better droplet monodispersity by reducing the droplet volume, or using serial electrodes that can improve sorting speed².”

4. While removing dead cells during encapsulation could be helpful, how does this compare with the typical strategy (sorting live cells before the 10x experiment)? Also, is it possible to FACS sort the droplet with encapsulated cells?

We thank the reviewer for their comment. One of the main benefits of spinDrop is that it collapses both sorting and single-cell encapsulation into a single experimental step. Therefore, we argue in the manuscript that this can be valuable for: 1) samples with low-input, where the combination of both sorting and single-cell encapsulation may prove too lossy in cell numbers, 2) samples with low viability, which may be suitable for sorting, but may still lyse in following processing step before encapsulation. 3) samples where known transcriptional signatures emerge through lengthy

dissociation and handling procedures, may be susceptible to significantly altered transcriptomes by the time encapsulation takes place. Our hope is that, through spinDrop, much of the experimental variability, which is instrument, personnel and sample dependent, can be removed, by sorting droplets directly at the moment of encapsulation; and would therefore supersede the combination of FACS and encapsulation.

Another advantage is the ability to threshold on signals with large areas and amplitudes (doublets) which enable superloading of the droplets (i.e. more than one cell every ten droplets). This largely supersedes current FACS and encapsulation methods in terms of handling throughputs and will be tremendously beneficial to cell atlasing efforts. In addition, even if the cell population loaded into encapsulation maintains high viability, empty droplets are still present in traditional workflows and contribute to background noise (via primer-dimer/concatemerization for example).

In addition, although the instrument cost for spinDrop (Supplementary table 10) is broadly similar to the 10x Chromium instrument, the added cost of FACS may prove cost-prohibitive for some sequencing centres. Furthermore, spinDrop could remove the variability in cell enrichment capabilities because of differing sensitivities across FACS machines or gating strategies.

Single emulsions cannot be processed in a flow cytometer because they rely on an aqueous sheath fluid as a carrier phase, whereas emulsions are only stable in an oil carrier phase. Double emulsions (water-in-oil-in-water²⁰) are generated by emulsifying water-in-oil droplets once more and appear overall aqueous, so that they are compatible with the aqueous sheath fluid carrier phase in FACS. They can be sorted using FACS, but we have not yet encountered this application in genomics, likely due to the difficulties of processing solid-support beads in the complex microfluidic and sorting workflows required for single-cell analyses. Ideally, droplets would become interfaced with FACS in the future, but the issue of multi-step processing remains, as no equivalent to a picoinjector currently exists for double emulsions. Furthermore, our microfluidic set-up is significantly less expensive than a FACS machine associated with a single-cell analysis instrument, therefore spinDrop could reach wider audiences that may require complex sample processing but do not have the infrastructure needed to convey these analyses.

5. In figure 3A, what is the heat denaturation step? More details are needed in the legend.

We agree that the figure merits further detailing to guide the reader better; we have now added the temperature and incubation time (70°C for 10 minutes) in the Figure caption (Figure R2.3) and in the text.

Figure R2.3 Figure 3A with clarifications regarding incubation temperature and time for the heat denaturation step.

6. How does the technique's efficiency compare with the latest 10x technique (v3)?

We agree with the reviewer that offering a comparison with the latest 10x Chromium v3 kit would have been more beneficial, but at the time of analysis, we did not have access to a HEK293T dataset using this kit. However, from the 10x application note, it appears the median genes detected for HEK293T cells with the 10x v3 system increases by 29.9% compared to the v2 chemistry (<https://kb.10xgenomics.com/hc/en-us/articles/360026501692-Do-we-see-a-difference-in-the-expression-profile-of-3-Single-Cell-v3-chemistry-compared-to-v2->

chemistry-). We have therefore clarified and underlined the most recent improvements from 10x Genomics in the text:

“10x Chromium has since then released a v3 formulation that increases median gene capture rates by 29.9% at maximum saturation on HEK293T cells according to their application note; further widening the gap between in-house and commercial scRNA-seq implementations. Therefore, further optimizations, such as utilising dissolvable beads for barcode release and fine-tuning barcode concentration⁸, could be implemented in the spinDrop workflow to further boost efficiency to match the latest 10x v3 improvements.”.

7. In Figure 3C, why do the mouse cells show relatively low purity? How does the purity compare with the 10x technique?

We agree that the whole-cell mouse ES and human HEK293T cells seem to indicate slightly lower purity for the mouse cells. After careful consideration, this might be explained by the following:

- 1) The human HEK293T cells were of lower viability when processing or dissociating the sample, leading to higher human reads in the droplets containing mouse ES cells. We believe this might be a reasonable explanation as the comparatively similar nuclei datasets, which rely less on viability during encapsulation, appear purer.
- 2) Higher capture in the spinDrop protocol leads to higher multi-mapping rates, as demonstrated by Ding *et al.*²¹ (and can be observed across methods in Figure R2.4a). The extent of cross species mapping may be dependent on protocol (as seen in R2.4c using the more sensitive 10xv3) and cell type.

We have computed the counts for species mixing experiments for the 10x v2 (Figure R2.4b) and v3 (Figure R2.4c) datasets using fresh frozen human HEK293T and mouse NIH3T3 cells. From this analysis, it appears that higher capture protocols (10x v3) may indeed lead to higher cross species mapping rates. For example, it seems the barcode counts relating to mouse (0-25% fraction) is less defined for the 10x v3 protocol, compared to 10x v2. This rejoins the results obtained with spinDrop, when applying similar lower thresholds for barcode selection ($n > 1,000$ genes per cell for either mouse or human genome; Figure R2.4d).

We have added a point of discussion in the text, explaining that our method does not perform well to remove ambient RNA (which would be the case in scenario 1) from sorted droplet, and a solution would be to lower the initial droplet volume:

“However, the approach does not remove ambient RNA from sorted droplets, hence methods like SoupX may still be required to remove ambient RNA co-encapsulated with cells from the dataset. Reduction of droplet size in the future (i.e. high cell volume to droplet volume ratio) may provide a viable route to remove or dampen such artefacts in future implementations.”

A) example from Ding et al.

B) 10x v2

C) 10x v3

D) spinDrop

Figure R2.4 (previous page). Comparative species-mixing experiments using 10x and other methodologies. A) Protocol-dependent species-mixing rates indicating higher crossover for more sensitive methods (for example CEL-seq2). Data taken from Ding et al. (Extended Data Fig 6b). B) Human/(Human+Mouse) UMI counts for a species mixing experiment using fresh frozen human HEK293T and mouse NIH3T3 cells with the 10x Chromium v2 protocol. C) Human/(Human+Mouse) UMI counts for a species mixing experiment using fresh frozen human HEK293T and mouse NIH3T3 cells with the 10x Chromium v3 protocol. D) Human/(Human+Mouse) UMI counts for a species mixing experiment using fresh frozen human HEK293T and mouse ES cells with the spinDrop protocol.

8. In Figure 4E, why the neural crest cells are missing in spinDrop? And the authors should compare the efficiency of the technique with 10x and sci-RNA-seq3 in this experiment.

We agree with the reviewer that neural crest cells appear depleted compared to the sciRNA-seq and 10x datasets, there are multiple non-mutually exclusive explanations to this.

First, we note that the samples used as an input may slightly differ depending on the dissection. For example, the sciRNA-seq3 dataset is a whole-embryo sample, as neural crest cells at E11.5 are found throughout the entire organism [REF], it may be that naturally this sample is more naturally enriched in neural crest cells. For the 10x Chromium dataset, the dissection mentions: “we isolated the entire cephalic part, including prospective forebrain, midbrain and hindbrain”, which resembles our dissection protocol. Therefore, it may be that capture of these cell types could be protocol-dependent. Indeed, benchmarking across sample types and single-cell methods have identified strong protocol-dependent biases in cell-type distributions²²(Figure R2.5a)

In addition, it may be that the neural crest themselves are more sensitive and were depleted in the input samples due to high cell death rates. Depletion of neural crest cells in scRNA-seq data from Neuroblastoma cells was also observed by Slyper *et al.*²³, however they were present in an equivalent nuclei extracted dataset, which

may signify these cell types are more prone to handling damage, and may explain their “disappearance” in our low viability sample (36.6% viability) (Figure R2.5d).

Of note, some neural crest cells are still observed in Cluster 8 as denoted by *Sox10* expression in Figure R2.5b-c. We further investigated the proportion of neural crest cells in our nascent 5EU-seq dataset. For this we took the non-nascent RNA fraction as cell-type proportions are not skewed by the analog’s diffusion and tissue-specific penetrance capabilities. After label-transfer, the proportion of neural crest cells in the sciRNA-seq3 dataset was 3.8% and 2.9% in spinDrop, which shows a slight depletion in our protocol.

In Figure R2.5d, we have plotted the distribution of the number of features for each dataset, demonstrating the superior performance of 10x compared to spinDrop dataset. spinDrop however outperformed sciRNA-seq3 in terms of captured genes per cell. However, it is worth mentioning that the datasets were not downsampled by matched sequencing coverage (unlike the benchmarking in Figure 2) as we used the datasets for data integration and label transfer mainly. In addition, the spinDrop datasets were not cut-off by a lower threshold for number of genes detected, which is not the case for the 10x dataset (“Cells with fewer than 2,000 UMIs were excluded from pooling”²⁴) or the sciRNA-seq3 dataset (“Cells with fewer than 200 UMIs or over 3,172 UMIs (two standard deviations above the mean UMI count) were discarded”²⁵). We have included these explanations in the discussion section, with the following text: “In our mouse brain analysis and nascent RNA analyses, neural crest cells were slightly depleted using spinDrop (2.9%) compared to sciRNA-seq3 (3.8%). Although neural crests have been shown to be enriched in nuclei data compared to whole-cell using a neuroblastoma clinical sample²³, further validation of selected droplets via imaging of cells being sorted would be beneficial to cell atlasing efforts.”

a) data by Ding et al.

b)

c)

d)

data by Silver et al.

Neuroblastoma (HTAPP-656-SMP-3481)

Cell type signature:

- B • Endothelial • Fibroblast • Macrophage • Neural crest • Neuroendocrine
- NK • T • Zona glomerulosa

a)

e)

Figure R2.5 (previous page). Neural crest depletion in the spinDrop dataset. a) data by Ding *et al.* showing that cell-type proportions vary across methods (in this case for PBMCs). b) UMAP representation of the clusters obtained for the mouse brain dataset at E10.5 with and without FADS, showing essentially similar cell-type proportions apart from fibroblasts (cluster 1). c) *Sox10* expression per cluster in the FADS-sorted sample. d) data by SLyper *et al.* showing that neural crest cells are depleted in neuroblastoma whole-cell scRNA-seq datasets, but not in its nuclei form, indicating that the neural crest are likely to be harder to lyse or more prone to cell death in whole-cell scRNA-seq workflows. e) number of genes detected per cell in the mouse brain dataset for spinDrop (with and without FADS), 10x Chromium and sciRNA-seq3.

9. How does the proportion of EU reads change across different cell types? This should be quantified.

We agree with the reviewer that exploring nascent RNA content per cell types may shed light on transcriptional dynamics in specific tissues. In the manuscript, we established the cell-type proportions that had both nascent and non-nascent RNA fractions and compared it to a ground-truth sciRNA-seq3 dataset to establish cell types with enriched proportional representation to characterise 5EU-analog tissue-specific uptake Extended Data Figure 5D. However, because the nascent RNA fractions can vary per cell-types and may indicate particular transcription kinetics, we have added this information to the newer version of the Figure, as can be seen in Figure R2.6. Broadly speaking, neural cell types were underrepresented in our analysis, perhaps informing on the tissue penetrance-specificity of the analog. This was also verified with the proportion of nascent RNA reads. One exception existed for the forebrain/midbrain, which displayed a larger proportion of nascent RNA counts (13%) than other neural tissues. Extra-embryonic tissues, endoderm tissues, blood and blood vessel cells were overall proportionally over-represented in our analysis; perhaps due to preferential diffusion of the analog as they are part of early circulation or part of the dissected extraembryonic tissues which may enhance analog diffusion. However, these tissues displayed average proportions of nascent counts. Overall, mesoderm cell types had higher levels of nascent RNA counts, especially the somatic mesoderm

and the heart fields indicating higher upticks in transcriptional activity. Of note, the allantois was removed from this analysis similarly to the data presented in Extended Data Figure 5D; there was an overlap from this cluster with low-complexity barcodes. Therefore we preferred to exclude this sample from the proportional representation analysis. We have added these elements of information in the text:

“Because the spinDrop dataset was filtered to contain barcodes represented both in the nascent and non-nascent libraries, proportions of cell types compared to the sciRNA-seq3 reference may inform on the capabilities of the analog to diffuse throughout the embryo. For example, primitive erythroids, gut, endothelium and extra-embryonic mesoderm and endoderm cells were proportionally enriched in the 5EU-seq dataset, whereas neural cell types such as the spinal cord, forebrain/midbrain and hindbrain were depleted (Extended Data Figure 5D). The fraction of nascent RNA reads was also smaller in tissues of neural origin, and higher in cell-types from mesodermal origin, in particular the somatic mesoderm, which displayed a high fraction of nascent RNA reads (24%) underlying tissue-specific transcriptional dynamics (Extended Data Figure 5D).”

Figure R2.6 Nascent RNA content across cell-types. The proportion of cells per cell-type with a matching barcode between nascent and non-nascent RNA generated using spinDrop is compared to the proportion per cell-type generated using an equivalent sciRNA-seq3 dataset ($\log_2(1+n)$ transformed on the x-axis), in function of nascent RNA fraction (y axis). Cell-types coloured by broad tissue classes.

10. It would be important that the authors comment on any potential limitations or concerns of the technique. Also, the author should provide detailed step-by-step protocol and computation script for the broad application of this technique.

We agree that further explanations on the potential drawbacks of the technique should be further provided, and have therefore added several new items in the discussion section as follows:

- 1) "The number of cells analyzed in this study (9,599) with spinDrop is in line with previous droplet proof-of-concepts (10,000 cells for inDrop¹⁴, 945 cells for DisCo-seq⁵⁷ and 498 cells for scRNA-seq combined with printed droplet microfluidics⁵⁸."
- 2) "10x Chromium has since then released a v3 formulation that increases median gene capture rates by 29.9% at maximum saturation on HEK293T cells according to their application note; further widening the gap between in-house and commercial scRNA-seq implementations. Therefore, further optimizations, such as utilising dissolvable beads for barcode release and fine-tuning barcode concentration⁸, could be implemented in the spinDrop workflow to further boost efficiency to match the latest 10x v3 improvements."
- 3) "However, the approach does not remove ambient RNA from sorted droplets, hence methods like SoupX may still be required to remove ambient RNA co-encapsulated with cells from the dataset. Reduction of droplet size in the future (i.e. high cell volume to droplet volume ratio) may provide a viable route to remove or dampen such artefacts in future implementations."
- 4) "Empty droplet removal in the spinDrop workflow will further enable counting of synthetic molecular RNA spike-ins (such as ERCC or Molecular spikes⁵⁹) along single-cell gene expression measurements. Indeed current workflows are constrained by Poissonian loading of single-cells into droplets; forcing high capture of synthetic RNA species in empty droplets which would increase sequencing costs significantly, largely explaining why this gold standard approach has not been transferred from plate-based to droplet-based assays. Removing empty droplets from the analysis circumvents these issues, and offers new avenues for removing counting biases from single-cell matrices in high-throughput datasets, drastically increasing the accuracy of any single-cell experimentation."

- 5) “Although sorting did not significantly skew cell-type proportions in our profiling endeavours, future implementations of the sorting technology could comprise an image analysis component to more faithfully discard cell doublets from large cells, or even add a phenomic component to the sequencing results, e.g. based on cell size or morphology.”
- 6) “Similarly, increased mitochondrial transcriptome coverage in spinDrop may benefit lineage tracing endeavours.”
- 7) “In our mouse brain analysis and nascent RNA analyses, neural crest cells were slightly depleted using spinDrop (2.9%) compared to sciRNA-seq3 (3.8%). Although neural crests have been shown to be enriched in nuclei data compared to whole-cell using a neuroblastoma clinical sample²³, further validation of selected droplets via imaging of cells being sorted would be beneficial to cell atlasing efforts. The throughput of the sorting might be increased by using smaller beads that would provide better droplet monodispersity, reducing the droplet volume, or using serial electrodes that can improve the speed of sorting². Further implementations may also include the use of dissolvable hydrogels⁸ or enzymatically released barcodes⁹ from the solid-support microgels to circumvent potential limitations of the current system, which uses UV-induced barcode release which may reduce RNA capture due to cross-linking.”

We have also added an experimental paragraph with detailed instructions that will enable the reader to replicate our protocol in the Methods section (“**Detailed protocol for the assembly the microfluidic rig and the spinDrop chip**”), including the scripts used for analysis and for sorting the droplets in LabVIEW: <https://github.com/droplet-lab/spinDrop> (to be deposited on GitHub). We also consolidated aspects of the workflow that were mentioned previously in separate sections into one narrative.

Reviewer #3 (Remarks to the Author):

spinDrop is proposed as an advancement for low cost, open source scRNA-seq with results comparable to that of 10X Genomics. In this technique, droplet sorting allows selection of those droplets that contain cells of interest, removing a significant amount of background noise from empty cells, dead cells, and other issues. This is followed by picoinjection to introduce reverse transcription (RT) reagents to select cells, allowing harsher lysis methods while still achieving RT in-droplet, similar to inDrop and 10x genomics protocols. Overall, the manuscript is straightforward. The authors present their technological additions, test them separately to confirm that they are achieving their goals (cell sorting/RT picoinjection), optimize parameters, and compare to other platforms.

I do not recommend this manuscript for publication as the work lacks novelty and does not seem to be of broad applicability.

We thank the reviewer for their feedback and note the main concern regarding novelty. Although it is true that several microfluidic architectures pertaining to cell and bead co-encapsulation, fluorescence-activated droplet sorting and picoinjection have been published, we believe our study present many new and unique aspects that should have a significant impact on the field of single-cell genomics; mainly performing bias-free single-cell analyses without inexact and costly computational data filtering, which has not been demonstrated previously. We have articulated our thoughts in the following bullet point list:

- spinDrop is the first in-house droplet microfluidic sorting set-up with significantly improved capture efficiency for the profiling of 3' mRNA. This is crucial to derive statistical power from single-cell experiments (for differential expression analysis for example). This alone promises to deliver single-cell RNA-sequencing analysis with comparable performance to state-of-the-art commercial methods, at a fraction of the cost (0.4 USD per cell for spinDrop, 1 USD per cell for the 10x Chromium).

- spinDrop is the first set-up to offer an elegant solution towards removing common biases found in droplet microfluidic such as empty droplets, dead cells and cell doublets. This considerably increases the accuracy of downstream analysis by removing the need for filtering the dataset with tools that show limited performance, but also significantly reduces sequencing cost by discarding the sources of bias before sequencing. We believe this is broadly transferable to other single-cell methodologies (such as ATAC-seq workflows, described using the Hydrop method⁸ for example).
- spinDrop is the first set-up to show high performance across input modalities which has not been demonstrated with similar microfluidic architectures: live cells, nuclei and PFA-fixed cells. This opens the door to a wide-range of previously unattainable tissues, such as fixed clinical samples or frozen brain nuclei samples. The availability of our method to process virtually any sample with demonstrated high quality will prove an invaluable asset in the clinic and for fundamental research.
- spinDrop processes PFA-fixed cells without the use of targeted RNA probes, in contrast to the 10x Genomics fixed RNA profiling kit. Therefore, our workflow enables applications where coding-sequences are important, such as expression quantitative trait *loci* characterizations using single-cell data²⁶. Or, for example, the characterization of species that are not mouse or human, which are the two only species characterizable using the single-cell fixed RNA sequencing kit from 10x.
- spinDrop is the first method demonstrating the capabilities to support synthetic RNA spike-ins (demonstrated here for ERCC). This application is usually not supported in droplet microfluidic formats, because the empty droplets would capture the synthetic molecules which would burden the sequencing cost significantly. Molecular spikes²⁷ have recently shown huge promise to correct biases in the gene expression matrix resulting in less biased counts.
- spinDrop shows performance across different staining types: calcein-AM for live cells, Vybrant green DNA stain for nuclei, fluorescently-tagged antibodies for cell-type sorting based on surface protein markers (such as B-cells). The versatility has not been demonstrated before in works that contain similar microfluidic architectures.

- spinDrop is the first method to show nascent RNA-sequencing using the 5EU-seq method in droplets, drastically increasing the resolution of differentiation trajectories.
- spinDrop is the first method to apply 5EU-seq to a whole-organism (nascent RNA-sequencing on E8.5 mouse embryos incubated with the 5EU analog.
- spinDrop is the first method to investigate differentiation kinetics using nascent RNA sequencing and contrast with velocity at a whole-organism scale
- spinDrop is the first method that employs sorting as a way to discard cellular multiplets to enable cell superloading. The gains in throughput associated to increasing cell loading fivefold have far-reaching implications for cell-atlasing experiments.
- The spinDrop method offers extensive benchmarking against other methods, which is not the case with other proof-of-concept microfluidic methods.

As for the lack of general applicability, we would politely like to refute this claim. Higher sensitivity, lower sequencing costs, lowered biases, cell-type specific sorting, competence in processing all input cell types and multi-step processing have applicability across virtually any high-throughput single-cell application, therefore our vision for where the field is heading does not align with these conservative statements. We have amended the manuscript according to this review using red lettering.

Major concerns:

1. The microfluidic schema proposed here is not conceptually new; see refs. 37, 58, <https://doi.org/10.3390/bioengineering9110674>

We agree that some sorting and pico-injector microfluidics architecture have been previously described. However we do not think that increased performance i.e. to boost sensitivity and reduce bias and sequencing costs, have been demonstrated before in the manner we describe in the previous bullet-points. We offer the following descriptions to clearly define the differences between spinDrop and the publications mentioned by the reviewer.

Reference 37 ⁵:

- VASA-seq: this methodology does not contain a sorter element, the core element of the spinDrop method, and therefore does not correct for the aforementioned biases and also does not provide for the massive reduction in sequencing costs that is one of the key deliverables of our present manuscript. Furthermore, VASA-seq was demonstrated only for whole-cells. Fixed cells (which are difficult to sequence in droplets) and nuclei, which are of important clinical significance, have not been evaluated using this method. Therefore our method demonstrates significant gains in the types of samples or protocols that can be run, with important clinical and translational underpinnings. In addition, no implementation for nascent RNA sequencing was described in that publication.

Reference 58 ²⁸:

- In this method, T-(Jurkat) and B- (Raji) cells are stained independently before being subjected to sorting: "*we apply the approach to a mixed population of B-cells (Raji) and T-cells (Jurkat), stained separately so they can be identified by their fluorescence*". We found multiple issues relating to the protocol, which would make their approach questionable when sampling a complex mixture like the one processed on our platform. First, the staining was not cell-specific, they used CellTrace Far Red and CellTrace Calcein to stain Raji and Jurkat cells separately. This is not applicable in a real-life scenario where different cell-types in a complex mixture can only be identified using cell-surface proteins. Furthermore, they mention in the text: "-90% B-cells (red, n = 1191) and -10% T- cells (green, n = 157)" were obtained after single-cell analysis, resulting in a total of 1,348 cells. However, the associated dimensional reduction plot in Figure 3C in their paper does only report on 1194 cells; this incompatibility perhaps raises questions on data analysis and how cells were filtered to appear on the plots. Therefore, no evidence exists for performance on a complex sample comprising multiple cell types, as described by our PBMC dataset, where there might be cell aggregates, co-encapsulation doublets, cross-talk between cell-types in terms of fluorescence signal or ambient background RNA. The performance of the sorter in these scenarios is thus questionable.

- In addition, no improvements were made to increase the sensitivity of RNA capture, and therefore there is no real improvement from the Drop-seq reaction conditions, which displays the worst performance (with inDrop) across benchmarking experiments^{21,22}.
- This method was only demonstrated for live cell sorting, which limits applicability to the cell-types we demonstrated in addition (fixed and nuclei). Furthermore, biases in the datasets and the potential for alleviating them were not explored nor quantified.
- Finally, there are many more distinct features in our method, for example the capabilities of sequencing nascent RNA or to process damaged samples or samples with synthetic RNA spike-ins. We believe our approach significantly improves on this works and provides more evidence of performance across modalities using reference datasets for benchmarking. Unfortunately, the work by Clarke *et al.* does not show any type of benchmarking, and it is therefore complicated to put it in context with other methods.

Last reference by Liu *et al.* 29:

We have carefully reviewed the article forwarded. However we found many caveats to the study, which we delineate in the following bullet points:

- The number of beads per cell varies widely for each droplet (multiple beads per cell) while they should be one or less. This would be problematic for downstream analysis as RNA molecules from single-cells will be split between barcodes (see Figure R3.1).
- They utilise incompressible barcoded beads, meaning that Poissonian loading applies to both cells and beads. The theoretical cell recovery rate should therefore be 1 %. Hence, the recovery rates stated by the authors have to be strongly influenced by bead doublets. Here, authors claim: “*Our study showed that the percentage of droplets without RNA-bead(s) was about 25% at the bead density of 25000/μL.*” The bead concentration of 25000/μL means that the average number of beads per droplet (lambda value) was around 1.2, assuming a 1:1 bead/cell flow ratio and 56.87 μm droplets, (Fig S3A) - corresponding to ~96.3 pL droplets. However, a lambda value of 1.2 would result in 30% empty droplets, and not 25% as observed by authors. This suggests that beads were

sedimented during the experiment and the average bead concentration in droplets was indeed larger. The 25% fraction of empty droplets translates to a lambda of ~ 1.4 which also means that more than 40% of droplets contain two or more beads and contribute to biased sequencing results. This 40% of droplets contain around 75% of the analysed cells (calculated using Poisson statistics). Therefore, due to the high percentage of droplets with double-beads, this method is not truly a single-cell RNA-seq method where a single-cell is expected to be paired to a single barcode and corresponding results should be evaluated with care.

- They do not evaluate the output of the unsorted channel. Therefore it is impossible to evaluate if sorting worked accurately (i.e. no viable single-cells are in the unsorted channel). Only a handful of droplets for the sorted channel are presented in Figure R3.1c, which is in addition contaminated by double-bead events as previously mentioned.
- In the Barnyard plot (Figure 4c), the dataset is truncated for human gene counts, which may skew cross-contamination calculations. A histogram plot of (human / (human + mouse) gene counts to observe the proportions of species-specific barcodes without thresholding should have been more informative for ultimately calculating multiplet rates.
- For comparative benchmarking experiments, no explanation on how downsampling normalisation between datasets was performed (and at which depth: reads per cell) was utilised for comparative analysis. Hence the analysis may be skewed by high sequencing coverage in their paper.
- In Figures 4 and 5, gene expression marker plots for the main cell types were absent so it is impossible to confidently claim that the population sorted is effectively T-cells. Again, to validate the sorter, the authors should have also provided sequencing results for the negative (unsorted) channel to demonstrate the sorter is in fact functional.
- no comments on (and indeed no analysis of) improvements in data quality and noise reduction are made, which we extensively discuss and demonstrate – and only these results would demonstrate the utility of a novel set-up.
- The analysis focuses on T-cell sorting and therefore does not demonstrate performance on broader atlases or cell-types.

- no demonstration of lower sequencing cost, lower bias, high performance for nuclei and fixed cells, cell-type specific sorting, molecular RNA spike-ins, comprehensive benchmarking against other methods.

a)

Supplementary figure S1B

b)

Supplementary Figure S6A

c) Figure 2C

Figure R3.1 Data from Liu *et al.* presenting results after encapsulation a), and sorting b) and c) red arrows illustrate a droplet failing to produce through single-cell data; green arrows indicate a single-cell co-encapsulated with a droplet.

2. The use of 2 different microfluidic devices make the workflow less user-friendly. The microfluidic driver setup is also complicated and unavailable in most laboratories. In particular, microfluidic picoinjectors and sorters are more difficult to fabricate than the passive droplet generators that are used for most droplet-based scRNA-seq (InDrop, Drop-seq, 10x Genomics). On-chip electrodes also have lower efficiency in fabrication electrical contact is easy to break during handling. Pairing two such devices make the workflow twice as likely to fail.

We believe there have been plenty of complex multi-step microfluidic platforms that have been published in which increased performance is demonstrated when compared to simpler passive microfluidic systems. We believe the complexity of multistep processing through multiple microfluidic steps to be necessary and constitutes the future of droplet single-cell technologies. As the number of modalities per single-cell increase (ATAC, RNA, Cut&Tag, proteomics etc.), the field will turn towards microfluidic systems like the ones described in this study to achieve complex successive handling steps on single-cell lysates (like the 5EU-seq read-out we propose). We agree with the reviewer that the complexity of implementation remains a barrier and have therefore significantly expanded the Methods section which should allow for implementation in less specialised laboratories.

We would like to also mention that commercial implementations of the sorter (e.g. STYX system from Atrandi Biosciences, <https://atrandi.com/styx>) and picoinjector or droplet fusion (Tapestri platform from MissionBio <https://missionbio.com/products/platform/> or ONYX system from Atrandi Biosciences, <https://atrandi.com/onyx>) exist, although they have not been implemented for our application (removing biases, increasing sensitivity and implementing multi-step protocols). Therefore, less specialised personnel may opt to work with these companies in the future, if they fail to implement our workflow.

The electrodes used in this study are a simple channel filled with a salt solution (introduced by Sciambi and Abate in 2014³⁰) and in our experience does not change experimental complexity or fabrication efficiency. Perhaps the reviewer refers to the molten alloy electrodes, which need alignment and make the assembly more complex, however this method is outdated.

3. The authors fail to discuss the throughput for the spinDrop setup: how many cells are processed in a typical experiment, using spinDrop? From fig. 4D, the cell numbers look to be too low to be of practical use (2,472 cells for spinDrop with FADS sorting and 802 cells for spinDrop without FADS sorting). I was unable to tell what the cell numbers were in Fig. 2E or Extended Data Figure 4A,B. Is this method scalable?

We touch on throughput in the discussion at page 18 of the manuscript but we have clarified our statement by adding following fragment in te methods sections: “... to generate droplets with average volume 1.3 ± 0.08 nL with a throughput of around 75 droplets per second that falls in a range of 10-100 Hz frequency of passive droplet generation in the original inDrop protocol¹⁴”. Additionally, the throughput of spinDrop is far superior to previously published single-cell applications combined with on-chip sorting (e.g. Bues *et al.*³¹ or Zhang *et al.*³²). In addition, we have demonstrated our system can perform well using 5 times the loading concentrations described in the system from Klein *et al.*¹⁷, by discarding doublets downstream at the sorting stage.

The number of cells analyzed is in line with previous droplet proof-of-concepts (10,000 cells for inDrop, 945 cells for DisCo-seq³¹ and 498 cells for scRNA-seq combined with printed droplet microfluidics³²; as compared to the 9,599 cells comprised in this study (Table R3.1 and Supplementary table 7 in the text). Although we appreciate the reviewer’s comments, they only mention one of the samples and omit the rest of the study which described many other samples. Although comprehensive atlasing for each sample (species mixing mouse ES and HEK293T cells, species -mixing nuclei mouse ES and HEK293T cells, fixed HEK293T cells, inDrop HEK293T cells with and without sorting, mouse brain at E8.5 with and without sorting, whole mouse embryo at E10.5 5EU-seq data, HEK293T with ERCC), would be preferred, we estimated the cost and time necessary to achieve large-scale datasets would not significantly ameliorate the proof-of-concept demonstrations proposed.

In the discussion we also added a description on how the sorting throughput can be increased in future iterations of the system: “Sorting throughputs might be increased by using smaller beads that would provide better droplet monodispersity by reducing the droplet volume, or using serial electrodes that can improve sorting speed⁶⁷”.

We have also added a point of discussion regarding cell numbers:

“The number of cells analyzed in this study (9,599) with spinDrop is in line with previous droplet proof-of-concepts (10,000 cells for inDrop¹⁴, 945 cells for DisCo-seq⁵⁷ and 498 cells for scRNA-seq combined with printed droplet microfluidics⁵⁸.”

4. Both droplet sorters and picoinjectors take longer to run (sort, picoinject) than passive droplet generators. What are the batch effects introduced from running ~10k cells in an experiment? Note that high cell numbers are typical in most single cell RNA-seq experiments (cell atlas’ing experiments).

Although picoinjection adds a microfluidic processing step, this step is performed on cell-containing sorted droplets (which can, in our experience, can be ran at 30-70 Hz), and is therefore overall much faster than the droplet generation/sorting microfluidic step, which has similar throughputs to other single-cell methods (75Hz, but on all droplets, including empty droplets). Therefore, the time required for picoinjection is negligible when put in the broader context of sample preparation, encapsulation, library preparation and sequencing; which can amount to several days of work/processing to obtain sample results. In addition, because our method does not require sample pre-enrichment (either through FACS or MACS), sample preparation times and batch effects due to lengthy sample processing procedures before lysis are greatly reduced using our method compared to other state-of-the-art high-throughput droplet methodologies.

We further agree that the goal of single-cell experiments is to cover cell-types of interest at high-enough coverage to derive statistical power to downstream analyses. However, for the purpose of the manuscript, we designed several experiments which did not substantially need extensive atlasing to demonstrate usefulness: 1) B-cells account for ~50% of the PBMC cells, hence sorting outputs can be reliably quantified with just a few hundred cells, 2) benchmarking experiments and optimisations also do require just a few 100 cells, there would not be a significant gain

in our analysis to sequence 10,000 HEK293T cells to derive conclusions in terms of RNA capture efficiency for example, 3) for the damaged mouse brain embryo sequencing sample, our aim was to show one could retrieve major cell types in a low viability and low cell number content sample (damaged brain sample from mouse embryos at E10.5). Although more cells could have been profiled in this case, our main focus was on showing higher coverage for viable cells compared to unsorted samples in this case. 4) for the nascent RNA sequencing experiment, the limited sample remaining after dissociation, fixing and filtering made it harder to obtain high cell numbers, but still the number was sufficient for a proof-of-principle nascent RNA sequencing across an entire embryo, and drive novel discoveries such as differential haemoglobin dynamics in differentiating primitive erythrocytes.

Type	Lymphoid cells sequencing (Fig 2.)	Type	HEK293 T benchmarking (Fig. 3)	Type	Mouse brain at E10.5 sorting (Fig.4)	Type	Mouse 5EU-seq data (Fig. 5)
Lymphoid sorted cells	157	All tested conditions	1,531	FACS sorted sample	2,472	Nascent RNA sequencing sample	936
Unsorted PBMCs	269	Species-mixing whole-cell	1,571	Unsorted sample	801		
		Species-mixing nuclei	412				
		Replicate HEK293T analysis	1210				
		PFA-fixed HEK293T	240				

Table R3.1 Number of cells obtained throughout different experiments throughout the manuscript.

Overall, our study comprises 9,599 cells across all conditions tested (Table R3.1), which is in line with other proof-of-concept single-cell genomics droplet studies comprising multiple sample types (inDrop¹⁷ study n= ~10,000 cells, DisCo-seq³¹ n=

~1,000 cells, Clark *et al.* sorter and merger ²⁸ n=5,001 cells). We anticipate future users not to be constrained by the throughput of the method. For example, follow-up inDrop manuscripts have sequenced significantly more cells than the initial story, and the encapsulation rate of our method does to significantly differ from inDrop. We have added Table R3.1 as a new Supplementary Table 7.

In any case, some experiments should be performed in replicates to demonstrate reproducibility.

We agree with the reviewer that measures of reproducibility would bolster our claims towards the broad applicability of the method. We therefore compared two separate HEK293T replicate samples processed with the spinDrop method. We first downsampled replicate 1 to match the sample size of replicate 2 (n= 610 cells) and compared the number of genes and UMIs per cell, which were broadly equivalent (replicate 1 median gene count per cell= 3,437.5, median UMI count per cell=5,786.5; replicate 2 median gene count per cell= 3,225, median UMI count per cell=5,430) (Figure R3.2a-b). We then processed the two replicate samples for dimensional reduction and projected them on the two first principal components without any batch effect correction. The two replicates spread homogeneously across both components, with no replicate-specific bias. We then proceeded to compare the average expression for both replicates, which had a R^2 of 0.98, further bolstering reproducibility claims. We have included these new materials in Figure 2C-D and Extended Data Figure 3F-G and added the following statements to the text:

"To test the reproducibility of spinDrop, two independent libraries prepared using HEK293T cells as an input were sequenced and analysed, showing similar gene and UMI capture per cell (Extended Data Figure 3F-G). The two replicates homogeneously spread across the two first principal components during dimensional reduction with no library-specific bias (Figure 2C) and correlation analysis of the average expression per gene showed high inter-replicate homology (R^2 =0.98, Figure 2D).".

a)

b)

c)

Figure R3.2
Replicate analysis of HEK 293T

d)

cells processed using the spinDrop method. a) violin plot representing the number of genes per cell between both replicates. b) violin plot representing the number of UMIs per cell between both replicates. c) PCA dimensionality reduction plot of both replicates. d) Correlation between the average expression for all genes in replicate 1 versus replicate 2.

Minor concerns:

1. When testing FADS, calcein-AM is used to get droplets with single, viable cells at a tested rate of (96.1%, n= 51). N=51 appears to be the single image presented in Figure 2B. Multiple images should have been processed for this test, including from different collections through FADs. A single image could be the result of an above average sorting run, is this 96.1% value consistent across multiple FADS run at different times?

We agree with the reviewer that additional sorting pictures could be provided to strengthen the claims of our sorter working robustly across sample input modalities

and experiments. Therefore we have added two additional sorting results in the manuscript pertaining to a sort with 5 times the typical cell loading concentration (to illustrate doublet removal, $\lambda = 0.5$) and a lymphoid cell types sorting using PE-labelled antibodies. From these additional data, we estimate the sorter to have reliably sorted 92.3% ($n=52$ droplets) and 95.3% ($n=43$ droplets) (Figure 3.3), which is in line with the claims from the manuscript. We have appended these new data to the manuscript in Extended Data Figure 2C.

Figure R3.3 Encapsulated cells sorted using different strategies with FADS. a) superimposed brightfield and blue light image of single-viable cells encapsulated in droplets after sorting, with an initial loading concentration of $\lambda = 0.5$. b) Red fluorescence image of sorted PBMC cells stained for lymphoid markers (CD19, CD45R and IgM) using PE-tagged antibodies at a concentration of $\lambda = 0.1$.

2. It is unclear why a species-mixing experiment was used to calculate the proportion of reads mapped to a cell barcode. What is the proportion of mapped reads in other spinDrop experiments, e.g., on embryonic mouse brain at E10.5 (Fig 4) and 5EU-seq (fig. 5)?

We apologise if this is unclear in the manuscript. Species-mixing experiments are used to validate that a single barcode represents a single-cell. This can be checked using a mixture of cells from two species as an input, and the number of reads mapping to each species after analysis informs on the relative cross-contamination rates. To

evaluate purity, we simply calculate the proportion, for each barcode, of reads mapping to the human over the mouse + human genome. In an ideal scenario, an accurate single-cell method will have, for each barcode, either a proportion of 100% (human) or 0% (mouse). This assay is a gold-standard assay to validate single-cell methodologies. This is quite different from mapping rates to the genome, which inform on the proportion of reads that are usable in downstream analysis post-mapping (i.e. to build the count matrix). We have calculated the number of mapped reads for the embryonic mouse brain at E10.5 (72.2%) and 5EU-seq (60.9% for the non-nascent part and 63.4% for the nascent RNA) and inserted these metrics in the Methods section.

3. It is unclear why single cells vs. single nuclei species mixing experiments yielded different doublet rates (2.9% for cells vs. 6.1% for nuclei, Extended Data Figure 4A, B). Was the target Poisson loading concentration different in the two experiments?

Because whole-cells and nuclei have different physical properties (buoyancy, stickiness, aggregation), different cross-contamination percentages can be obtained despite similar cell loading rates. In our experiments, we aimed to obtain ~5% of species-mixing; we believe the values reported here do not significantly diverge from this and are globally in line with other droplet-based single-cell platforms.

4. The Wilcoxon rank sum test (lines 284-288) result could be made clearer. Out of the 690 differentially expressed genes, how many were upregulated in spinDrop vs 10x? What does this mean?

We apologise for the lack of clarity in the sentence. We detected 690 genes that were robustly differentially expressed during our comparison between 10x and spinDrop. Of this total number of 690 genes, 291 genes were overexpressed in spinDrop, 399 genes were overexpressed in the 10x dataset. We have modified the text accordingly, for clarity:

“The analysis revealed a total of 690 genes were significantly and robustly differentially expressed throughout the dataset (absolute values for log₂ fold change > 1 and Bonferroni adjusted p-values <10⁻⁵). Further annotation of the genes by

biotypes showed an enrichment for non-coding RNAs and pseudogenes for spinDrop and some protein-coding genes in the 10x Chromium dataset (Extended Data Figure 3D, Supplementary Table 2). From the list of significantly differentially expressed genes, spinDrop showed an enrichment of 291 genes which, classified proportionally per biotype, were: 1) 2.1% lncRNAs, 2) 26.4% processed pseudogenes, 3) 63.9% protein-coding, 4) 2.7% transcribed processed pseudogenes and 5) 2.7% unprocessed pseudogenes. The 10x dataset, on the other hand, had 399 genes that were upregulated which, classified proportionally per biotype, were: 1) 99% protein-coding genes and 2) 1% of lncRNAs.”

References

1. Garstecki, P., Stone, H. A. & Whitesides, G. M. Mechanism for flow-rate controlled breakup in confined geometries: a route to monodisperse emulsions. *Phys. Rev. Lett.* **94**, 164501 (2005).
2. Isozaki, A. *et al.* Sequentially addressable dielectrophoretic array for high-throughput sorting of large-volume biological compartments. *Sci Adv* **6**, eaba6712 (2020).
3. Sciambi, A. & Abate, A. R. Accurate microfluidic sorting of droplets at 30 kHz. *Lab Chip* **15**, 47–51 (2015).
4. Sheth, R. U. *et al.* Spatial metagenomic characterization of microbial biogeography in the gut. *Nat. Biotechnol.* **37**, 877–883 (2019).
5. Salmen, F. *et al.* High-throughput total RNA sequencing in single cells using VASA-seq. *Nat. Biotechnol.* (2022) doi:10.1038/s41587-022-01361-8.
6. Hafner, M. *et al.* CLIP and complementary methods. *Nature Reviews Methods Primers* **1**, 1–23 (2021).
7. Harris, M. E. & Christian, E. L. RNA crosslinking methods. *Methods Enzymol.* **468**, 127–146 (2009).
8. De Rop, F. V. *et al.* Hydrop enables droplet-based single-cell ATAC-seq and single-cell RNA-seq using dissolvable hydrogel beads. *Elife* **11**, e73971 (2022).
9. Delley, C. L. & Abate, A. R. Modular barcode beads for microfluidic single cell genomics. *Sci. Rep.* **11**, 10857 (2021).
10. Young, M. D. & Behjati, S. SoupX removes ambient RNA contamination from droplet-based single-cell RNA sequencing data. *Gigascience* **9**, giaa151 (2020).
11. Vallejo, A. F. *et al.* snPATHO-seq: unlocking the FFPE archives for single nucleus RNA profiling. *bioRxiv* 2022.08.23.505054 (2022) doi:10.1101/2022.08.23.505054.
12. Lin, L. *et al.* LINEAGE: Label-free identification of endogenous informative single-cell mitochondrial RNA mutation for lineage analysis. *Proceedings of the National Academy of Sciences* **119**, e2119767119 (2022).
13. Argelaguet, R. *et al.* Multi-Omics Factor Analysis—a framework for unsupervised integration of multi-omics data sets. *Mol. Syst. Biol.* **14**, e8124 (2018).
14. Bergen, V., Lange, M., Peidli, S., Wolf, F. A. & Theis, F. J. Generalizing RNA velocity to transient cell states through dynamical modeling. *Nat. Biotechnol.* **38**,

- 1408–1414 (2020).
15. Zatulovskiy, E. & Skotheim, J. M. On the Molecular Mechanisms Regulating Animal Cell Size Homeostasis. *Trends Genet.* **36**, 360–372 (2020).
 16. Ginzberg, M. B., Kafri, R. & Kirschner, M. On being the right (cell) size. *Science* **348**, 1245075 (2015).
 17. Klein, A. M. *et al.* Droplet Barcoding for Single-Cell Transcriptomics Applied to Embryonic Stem Cells. *Cell* **161**, 1187–1201 (2015).
 18. Colin, P.-Y. *et al.* Ultrahigh-throughput discovery of promiscuous enzymes by picodroplet functional metagenomics. *Nat. Commun.* **6**, 1–12 (2015).
 19. Schnettler, J. D., Klein, O. J., Kaminski, T. S., Colin, P.-Y. & Hollfelder, F. Ultrahigh-Throughput Directed Evolution of a Metal-Free α/β -Hydrolase with a Cys-His-Asp Triad into an Efficient Phosphotriesterase. *J. Am. Chem. Soc.* **145**, 1083–1096 (2023).
 20. Zinchenko, A. *et al.* One in a Million: Flow Cytometric Sorting of Single Cell-Lysate Assays in Monodisperse Picolitre Double Emulsion Droplets for Directed Evolution. *Anal. Chem.* **86**, 2526–2533 (2014).
 21. Ding, J. *et al.* Systematic comparison of single-cell and single-nucleus RNA-sequencing methods. *Nat. Biotechnol.* **38**, 737–746 (2020).
 22. Mereu, E. *et al.* Benchmarking single-cell RNA-sequencing protocols for cell atlas projects. *Nat. Biotechnol.* **38**, 747–755 (2020).
 23. Slyper, M. *et al.* A single-cell and single-nucleus RNA-Seq toolbox for fresh and frozen human tumors. *Nat. Med.* **26**, 792–802 (2020).
 24. La Manno, G. *et al.* Molecular architecture of the developing mouse brain. *Nature* **596**, 92–96 (2021).
 25. Cao, J. *et al.* The single-cell transcriptional landscape of mammalian organogenesis. *Nature* **566**, 496–502 (2019).
 26. Nathan, A. *et al.* Single-cell eQTL models reveal dynamic T cell state dependence of disease loci. *Nature* **606**, 120–128 (2022).
 27. Ziegenhain, C., Hendriks, G.-J., Hagemann-Jensen, M. & Sandberg, R. Molecular spikes: a gold standard for single-cell RNA counting. *Nat. Methods* **19**, 560–566 (2022).
 28. Clark, I. C. *et al.* Targeted Single-Cell RNA and DNA Sequencing With Fluorescence-Activated Droplet Merger. *Anal. Chem.* **92**, 14616–14623 (2020).
 29. Liu, Y. *et al.* Droplet Microfluidics Enables Tracing of Target Cells at the Single-Cell Transcriptome Resolution. *Bioengineering (Basel)* **9**, (2022).
 30. Sciambi, A. & Abate, A. R. Generating electric fields in PDMS microfluidic devices with salt water electrodes. *Lab Chip* **14**, 2605–2609 (2014).
 31. Bues, J. *et al.* Deterministic scRNA-seq captures variation in intestinal crypt and organoid composition. *Nat. Methods* **19**, 323–330 (2022).
 32. Zhang, J. Q. *et al.* Linked optical and gene expression profiling of single cells at high-throughput. *Genome Biol.* **21**, 49 (2020).

REVIEWERS' COMMENTS

Reviewer #1 (Remarks to the Author):

This paper has several technical accolades in the scRNA-seq field ranging from sorting cells successful in co-encapsulation for viability, reagent addition via dielectrophoretic picoinjection to maintain small volumes, and demonstrating the advantages of their methods using excellent analyses. The significance of their work extends beyond just scRNA-seq and no doubt will be widely applicable in several branches of microfluidics. The authors have fully addressed my comments, thank you. The methodology is now reproducible, clarified, and sound. The manuscript is ready for publication.

Reviewer #2 (Remarks to the Author):

The authors have effectively addressed all my concerns. I commend the authors for their quality control analysis, comprehensive comparisons, and enlightening discussions. There is no additional feedback regarding the manuscript, and I am excited about this study's contribution to the field of single-cell genomics.

Reviewer #3 (Remarks to the Author):

After careful consideration of the authors' response to my original assessment, I do not recommend this manuscript for publication. While the work is executed well and the authors have made some good points on the novel aspects, I am not convinced that the method will be of broad use to the community, given my concerns with throughput vs. difficulty in execution.

Responses to the reviewers' comments for manuscript NCOMMS-23-00685-T

Last changes to the text of the main paper are shown in red in the submitted copy.

Reviewer #1:

This paper has several technical accolades in the scRNA-seq field ranging from sorting cells successful in co-encapsulation for viability, reagent addition via dielectrophoretic picoinjection to maintain small volumes, and demonstrating the advantages of their methods using excellent analyses. The significance of their work extends beyond just scRNA-seq and no doubt will be widely applicable in several branches of microfluidics. The authors have fully addressed my comments, thank you. The methodology is now reproducible, clarified, and sound. The manuscript is ready for publication.

We thank the reviewer for the constructive review process and are thrilled to have incorporated their suggestions in the current version of our manuscript.

Reviewer #2:

The authors have effectively addressed all my concerns. I commend the authors for their quality control analysis, comprehensive comparisons, and enlightening discussions. There is no additional feedback regarding the manuscript, and I am excited about this study's contribution to the field of single-cell genomics.

We thank the reviewer for their helpful comments and for helping us to strengthen our manuscript. We are excited to share spinDrop with the single-cell community.

Reviewer #3:

After careful consideration of the authors' response to my original assessment, I do not recommend this manuscript for publication. While the work is executed well and the authors have made some good points on the novel aspects, I am not convinced that the method will be of broad use to the community, given my concerns with throughput vs. difficulty in execution.

We thank the reviewer for acknowledging the technical quality of the work and its novelty, revising his/her previous doubts about the latter. The additional point that the reviewer has made in the second round of comments is well taken: the ease of routine implementation, including in non-expert labs, is crucial for the success of a method in a cross-disciplinary endeavour. Reviewer #1 judges that the preconditions for wide uptake and implementation are in place: *"The methodology is now reproducible, clarified, and sound"*. Reviewer #2 emphasises the attractiveness of our *"quality control analysis, comprehensive comparisons, and enlightening discussions"* as drivers of future adaptation. These judgements augur well for a *"broad use to the*

community”. The ultimate test is the experimental reality, but publication is the necessary for dissemination before community uptake can be assessed.

We are confident that our efforts to strengthen the Methods section in the previous round of revision will be sufficient to ensure that procedures and implemented reproducibly, as underlined by the two other reviewer’s comments.

For users with without prior microfluidics experience we have now added suggestions for potential commercial suppliers to the Methods section of our manuscript: Atrandi Biosciences via the Styx instrument for droplet generation and sorting, and the Onyx system for picoinjection. These will equip researchers with ‘push-button instrumentation’. Accordingly we have added: “*Alternative commercial instrumentation may also be employed to achieve FADS and picoinjection using push-button solutions, such as the Styx (FADS) and Onyx (picoinjection) from Atrandi Biosciences.*”

We do not believe throughput is a limitation in our workflow. Indeed, our study boasts ~10,000 cells, which is in line or vastly exceeds similar efforts reported in recent publications (Supplementary Table 7). In an effort to increase adoption by the community we show proof-of-concept data for multiple applications, to reassure users who may want to review examples before implementing our microfluidic solution to suit their specific needs. To this end, we have prioritised the profiling of numerous cell types under different experimental scenarios (species mixing mouse ES and HEK293T cells, species -mixing nuclei mouse ES and HEK293T cells, fixed HEK293T cells, inDrop HEK293T cells with and without sorting, mouse brain at E8.5 with and without sorting, whole mouse embryo at E10.5 5EU-seq data, HEK293T with ERCC, PBMC lymphoid cell-type sorting) rather than one sample with many cells. Furthermore, each experiment was designed so that each cell type could be covered sufficiently to ascertain sorting output composition and provide lists of unique molecular markers that were uncovered using spinDrop; we have explained the reasoning for selecting specific sample sizes in the reporting summary. The throughput of the FADS workflow is 75 Hz, which is in line with other droplet microfluidic single-cell solutions, as explained in the previous response. As for the picoinjection, this step is performed uniquely on sorted cells (and not empty droplets) at 30-70 Hz (up to 252,000 cells per hour processed). We do not believe that this step is significantly time-constraining when put in the broader context of cell culture, dissociation, preparation, encapsulation, library preparation, sequencing and data analysis. However, we agree that this should also be stated in the current version of the manuscript, therefore we have added the following statement to the discussion: “Although spinDrop requires a picoinjection step after cell sorting, the throughput of this microfluidic handling step is manageable (up to 70 Hz, equivalent to 252,000 cells/hour). Furthermore, at this stage empty droplets had already been discarded during sorting. As a consequence, the picoinjection operation is exclusively performed on droplets containing cell lysates, which means the demands in terms of throughput for this step are substantially lower than for the initial droplet formation.”

”

In addition, we have made video recordings of the sorting and picoinjection steps available as Supplementary Videos 1 and 2 that will enable readers to confirm our claims regarding throughput.

As for the limited applicability and broad-usage of spinDrop, we are encouraged by the the two other reviews and would like to reemphasise, in response to reviewer #3, the potential of our method to transform single-cell research. Higher quality data provided at lower cost is at the bleeding edge of scRNA-seq method development, as they are the key to constructing the superior datasets (both larger size and higher quality) that are necessary to further our understanding of biology. We maintain that our approach offers an elegant solution to these two main bottlenecks in single-cell research; and will therefore naturally encourage adoption by the community.